# Pyruvic acid in the boreal forest: Gas-phase mixing ratios and impact on radical chemistry

Philipp G. Eger[1], Jan Schuladen[1], Nicolas Sobanski[1], Horst Fischer[1], Einar Karu[1], Jonathan Williams[1], Matthieu Riva[2,3], Qiaozhi Zha[2], Mikael Ehn[2], Lauriane L. J. Quéléver[2], Simon Schallhart[2], Jos Lelieveld[1], and John N. Crowley[1]

[1]Atmospheric Chemistry Department, Max-Planck-Institute for Chemistry, 55128 Mainz, Germany
[2]Institute for Atmospheric and Earth System Research / Physics, FI-00014 University of Helsinki, Finland
[3]University of Lyon, Université Claude Bernard Lyon 1, CNRS, IRCELYON, F-69626, Villeurbanne, France

*Correspondence to*: John N. Crowley (john.crowley@mpic.de)

**Abstract.** Pyruvic acid, $CH_3C(O)C(O)OH$, is an organic acid of biogenic origin that plays a crucial role in plant metabolism, is present in tropospheric air in both gas-phase and aerosol-phase and is implicated in the formation of secondary organic aerosols (SOA). Up to now, only a few field studies have reported mixing ratios of gas-phase pyruvic acid and its tropospheric sources and sinks are poorly constrained. We present the first measurements of gas-phase pyruvic acid in the boreal forest as part of the IBAIRN (Influence of Biosphere–Atmosphere Interactions on the Reactive Nitrogen budget) field campaign in Hyytiälä, Finland, in September 2016. The mean pyruvic acid mixing ratio during IBAIRN was 96 pptv, with a maximum value of 327 pptv. From our measurements we estimated the overall pyruvic acid source strength and quantified the contributions of isoprene oxidation and direct emissions from vegetation in this monoterpene-dominated, forested environment. Further, we discuss the relevance of gas-phase pyruvic acid for atmospheric chemistry by investigating the impact of its photolysis on acetaldehyde and peroxy radical production rates. Our results show that, based on our present understanding of its photo-chemistry, pyruvic acid is an important source of acetaldehyde in the boreal environment, exceeding ethane/propane oxidation by factors of ~ 10 and ~ 20.

## 1 Introduction

Organic acids play a crucial role in tropospheric chemistry. They influence the acidity of aerosols and cloud droplets and are involved in the formation of secondary organic aerosol (SOA), thereby impacting air quality and climate (Kanakidou et al., 2005; Hallquist et al., 2009). Pyruvic acid ($CH_3C(O)C(O)OH$), the simplest α-keto-acid, is omnipresent in plants where it is central to the metabolism of isoprene, monoterpenes and sesquiterpenes (Eisenreich et al., 2001; Magel et al., 2006; Jardine et al., 2010) and is also found in tropospheric air, especially in the boundary layer of vegetated regions (see Sect. 1.3).

The boreal forest is one of the largest terrestrial biomes on Earth covering about 10 % of its land surface and emitting large amounts of biogenic VOCs into the atmosphere (Kesselmeier and Staudt, 1999; Rinne et al., 2005; Hakola et al., 2012). It

serves as an important global carbon reservoir (Bradshaw and Warkentin, 2015) and impacts the Earth's climate not only through forest-atmosphere carbon exchange but also via surface albedo, evapotranspiration and formation of cloud condensation nuclei and SOA from gaseous biogenic precursors (Kulmala et al., 2004; Bonan, 2008; Sihto et al., 2011). Our work focusses on the first measurement and chemical impact of gas-phase pyruvic acid in a boreal forest environment.

## 1.1 Atmospheric sources of pyruvic acid

There are several known routes to the photochemical formation of gas-phase pyruvic acid in the troposphere. In clean air, pyruvic acid is generated during the photo-oxidation of isoprene via the ozonolysis of methylvinylketone (MVK) and subsequent hydrolysis of the Criegee intermediates formed (Jacob and Wofsy, 1988; Grosjean et al., 1993; Paulot et al., 2009). Pyruvic acid is found in the photolysis (in air) of methylglyoxal (Raber and Moortgat, 1995), itself formed from the OH-initiated oxidation of several biogenic VOCs (Arey et al., 2009; Obermeyer et al., 2009) including monoterpenes (Yu et al., 1998; Fick et al., 2003). Pyruvic acid is also formed in the reactions of peroxy radicals generated in the oxidation of propane, acetone and hydroxyacetone (Jenkin et al., 1993; Warneck, 2005) and in the gas-phase photo-oxidation of aromatics in the presence of $NO_X$ (Grosjean, 1984; Praplan et al., 2014).

In the condensed phase, the aqueous-phase oxidation of methylglyoxal leads to the formation of pyruvic acid (Stefan and Bolton, 1999). Biomass burning also results in the formation of pyruvic acid (Andreae et al., 1987; Helas et al., 1992) where the heterogeneous photooxidation of particle-phase aromatics plays a role (Pillar et al., 2014; Pillar et al., 2015; Pillar and Guzman, 2017; Pillar-Little and Guzman, 2018). The latter also results in formation of a further oxo-carboxylic acid with the same molecular mass (3-oxo-propionic acid). Finally, pyruvic acid is believed to be directly emitted by vegetation as indicated by measurements of very high mixing ratios under oxidation-free conditions in a tropical rain-forest biome (Jardine et al., 2010).

## 1.2 Atmospheric sinks of gas-phase pyruvic acid

Like other di-carbonyls, pyruvic acid has a UV-absorption spectrum that extends into the visible part of the electromagnetic spectrum (Horowitz et al., 2001) and is thus photolysed rapidly by actinic radiation. Experimental studies indicate that, in the gas-phase, pyruvic acid has a lifetime with respect to photolysis of a few hours (Grosjean, 1983; Winterhalter et al., 2001). In contrast, the rate constant for reaction of pyruvic acid with OH is slow ($1.2 \times 10^{-13}$ $cm^3$ molecule$^{-1}$ s$^{-1}$, IUPAC (2019)) and this may be considered a negligible sink with a lifetime of ~ 3 months (Mellouki and Mu, 2003). The photolysis of pyruvic acid proceeds mainly (~ 60%) via exothermic decarboxylation involving a 5-membered transition state that decomposes to $CO_2$ and methyl-hydroxycarbene ($CH_3COH$), the latter rearranging to acetaldehyde (IUPAC, 2019). Other product channels observed are $CH_3CO + HOCO$ ~ 35 %) and $CO + CH_3C(O)OH$ (~ 5 %) (see section 3.2 for more details). With a Henry's law constant of ~ $3 \times 10^5$ M atm$^{-1}$ (Staudinger and Roberts, 1996), pyruvic acid is highly soluble (due to formation of a germinal diol, (Guzman et al., 2006)) and wet and dry deposition and partitioning into the aerosol-phase are expected to be important sinks, especially at high relative humidity, thus contributing to SOA formation (Carlton et al., 2006; Tan et al.,

2012; Griffith et al., 2013; Reed Harris et al., 2014; Eugene and Guzman, 2017; Eugene and Guzman, 2019; Mekic et al., 2019).

### 1.3 Observations of ambient, gas-phase, pyruvic acid

Pyruvic acid was first observed by Andreae et al. (1987) in the Amazonas region (Brazil) as well as in the southern US, with most (85–93%) found in the gas-phase, where mixing ratios ranged from 10 to 400 pptv. Andreae et al. (1987) reported the highest mixing ratios for the Amazon forest near the top of the forest canopy, which was considered consistent with formation from the oxidation of isoprene in the boundary layer and removal by dry deposition. Similarly, average daytime mixing ratios of pyruvic acid over central Amazonia of $(25 \pm 15)$ pptv (forest canopy) and $(15 \pm 15)$ pptv (free troposphere) (Talbot et al., 1990) were consistent with model predictions (Jacob and Wofsy, 1988) of pyruvic acid formation from isoprene degradation. Helas et al. (1992) found pyruvic acid mixing ratios up to 800 pptv in and above the equatorial African rain forest which could not be attributed to isoprene oxidation, indicating additional sources. Pyruvic acid levels of up to 200 pptv in the rural continental atmosphere at a mountain top site over the Eastern U.S. were thought to originate from biogenic emissions and possibly photochemical production (Talbot et al., 1995). In regions influenced by anthropogenic emissions, pyruvic acid has been measured at mixing ratios of up to 500 pptv whereby the diurnal profiles indicated a dominant photochemical source (Mattila et al., 2018), and it was present in an urban air mass in the Los Angeles Basin and New York (Khwaja, 1995; Veres et al., 2008). Very low mixing ratios ($\sim$ 1 pptv) of pyruvic acid were found in the marine boundary layer over the Atlantic Ocean (63 °N to 39 °S), confirming the importance of continental sources (Baboukas et al., 2000). Mixing ratios of pyruvic acid (up to 15 ppbv) were reported in an experimental tropical rain forest enclosure (Jardine et al., 2010) and were accompanied by isoprene levels exceeding 100 ppbv with other terpenoids up to $\sim$ 10 ppbv. In the enclosure, photochemical production and loss of pyruvic acid are not important and the high pyruvic acid mixing ratios were attributed to direct emissions.

Pyruvic acid is a potentially important but unexplored atmospheric component which is present in the gas-phase as well as in the aerosol phase (Andreae et al., 1987) and, along with other di-carbonyls, has been proposed to be a potentially important source of $CH_3C(O)O_2$ and $HO_2$ radicals in areas dominated by biogenic emissions (Crowley et al., 2018). So far, elevated pyruvic acid mixing ratios have only been observed in temperate or equatorial forests where isoprene emissions were large. In the following, we present the first gas-phase measurements of pyruvic acid in the boreal forest where isoprene levels (in September) were generally low and investigate its impact on photochemical radical production in this environment.

### 2 Methods

The IBAIRN campaign (Influence of Biosphere–Atmosphere Interactions on the Reactive Nitrogen budget) took place in the boreal forest in Hyytiälä, Finland, in September 2016 during the summer-autumn transition. Measurements were performed at the "Station for Measuring Forest Ecosystem-Atmosphere Relations II" (SMEAR II) in Hyytiälä (61.846 °N, 24.295 °E,

180 m above sea level) in southern Finland (Hari and Kulmala, 2005), in a forested area which is characterised by large biogenic emissions and low $NO_x$ concentrations (Williams et al., 2011; Crowley et al., 2018; Liebmann et al., 2018). The vegetation in the surrounding 50 km is dominated by Scots pine and Norway spruce and the site is only occasionally influenced by anthropogenic emissions, with the nearest city (Tampere) located $\approx$ 50 km to the south-west. A detailed

description of the measurement site can be found elsewhere (Hari and Kulmala, 2005; Hari et al., 2013). Meteorological parameters including wind direction, wind speed, temperature, relative humidity and precipitation are continuously monitored at various heights on the 128 m SMEAR II tower and distributed via an on-line data exploration and visualisation tool for SMEAR stations (Junninen et al., 2009). Measurements of $NO_3$ radical reactivity, alkyl nitrates, highly oxygenated molecules (HOM) and meteorological parameters during the IBAIRN campaign have recently been reported (Liebmann et

al., 2018; Zha et al., 2018; Liebmann et al., 2019). Unless stated otherwise, the trace-gases discussed in this paper were sampled from the centre of a high volume-flow inlet (10 $m^3 min^{-1}$, 0.15 m diameter, 0.2 s residence time) made of stainless steel, the top of which was located at a height of 8 m above the ground. The top of the canopy around the clearing was at ~ 20 m.

**2.1 CI-QMS measurement of pyruvic acid**

Pyruvic acid was detected with a chemical ionisation quadrupole mass spectrometer (CI-QMS) equipped with an electrical, radio-frequency (RF) discharge ion-source, described in detail by Eger et al. (2019). The CI-QMS detected pyruvic acid as $CH_3C(O)C(O)O^-$ at a mass-to-charge ratio ($m/z$) of 87. The sensitivity was 4.8 Hz per pptv of pyruvic acid for a (typical) primary-ion count rate (at $m/z = $ 127) ~$10^6$ Hz.

which resulted in a detection limit (LOD) of 15 pptv (10 s, 2$\sigma$) or 4 pptv (10 min). The detection scheme is believed to be

similar to the one reported for acetic acid (Eger et al., 2019) and involves the reaction of pyruvic acid with $I(CN)_2^-$ primary ions to initially form HCN + $I(CN)CH_3C(O)C(O)O^-$ ($m/z$ 240) which then dissociates to $CH_3C(O)C(O)O^-$ ($m/z$ 87) when a 20 V de-clustering voltage is applied in the collisional dissociation chamber. The $I(CN)_2^-$ ion was not monitored continuously during the IBAIRN campaign and the signal at $m/z$ 87 was converted to a mixing ratio after normalisation to the ion count of the major primary ion, $I^-$. As the I and C atoms in $I(CN)_2^-$ stem from $CH_3I$, we expect the concentration of $I(CN)_2^-$ to be

proportional to that of $I^-$ (which was monitored continuously).

As detection of pyruvic acid during IBAIRN was not expected, the instrument was calibrated post-campaign by simultaneously monitoring the output of a diffusion source (98% pyruvic acid, Sigma-Aldrich) with the CI-QMS and an infrared absorption spectrometer measuring $CO_2$ (LI-COR) following the complete oxidation of pyruvic acid to 3 $CO_2$ molecules in air, using a palladium catalyst at 350 °C (Veres et al., 2010). A calibration curve is given in Fig S1 of the

supplementary information. The CI-QMS sensitivity to pyruvic acid was found to be independent of relative humidity (RH) for RH > 20 %. In dry air the sensitivity drops to about 60 % of that observed with humidified air reflecting the importance of water clusters in the reaction with the primary ion.

A flow of 2.5 L (standard temperature and pressure, STP) min$^{-1}$ was drawn into the CI-QMS via a 3 m long 6.35 mm (outer diameter, OD) PFA tubing and then a 20 cm section of PFA that was heated to 200 °C, required for PAN detection (Eger et al., 2019). During calibration, we found no change in the pyruvic acid signal when the inlet was at either room temperature or heated to 200 °C. We cannot completely rule out that some unknown secondary reactions at 200 °C may influence the pyruvic acid concentration during ambient measurements, though given the short inlet residence time (200 ms), we consider this to be unlikely.

A membrane filter (Pall Teflo, 2 µm pore) was placed between the high-volume inlet and CI-QMS sampling line to remove particles and was exchanged regularly to avoid accumulation of particulate matter. The ion molecule reactor was held at a pressure of 18.00 mbar (1 mbar = 100 Pa) and a de-clustering voltage of 20 V was applied in the collisional dissociation chamber.

The background signal at all masses monitored was determined by periodically passing ambient air (for 10 min) through a scrubber filled with steel wool where pyruvic acid was efficiently destroyed at the hot surfaces (120 °C). Owing to pyruvic acid's high affinity for surfaces, even after 5–10 minutes of scrubbing, the signal did not go to zero (Fig. S2), which resulted in a background signal that co-varied with the ambient signal at $m/z$ 87. This is illustrated in Fig. S2 of the supplementary information in which we show the raw signal at $m/z$ 87 and the signal at the same mass when sampling via the scrubber. The background signal for $m/z$ 87 (red dashed line in upper panel Fig. S2) was therefore determined from measurements in which pyruvic acid mixing ratios were close to the detection limit during the early part of the campaign. This choice could be confirmed by examining the background signal when the inlet was overflowed with zero-air. We have increased the total uncertainty on the pyruvic acid mixing ratios to 30% (of the mixing ratio) ± 20 pptv to take account of this.

The sensitivity of the CI-QMS to pyruvic acid can be accurately derived from laboratory-based calibrations. However, $m/z$ 87 is subject to potential interferences owing to the limited mass resolution (~ 1 atomic mass unit, amu) of our quadrupole mass spectrometer. In the following, we discuss potential contributions of other trace gases to $m/z$ 87 and examine the evidence that supports the assignment to (predominantly) pyruvic acid.

Analogous to the detection of PAN (CH$_3$C(O)O$_2$NO$_2$) as the acetate anion at $m/z$ 59, we would expect the CI-QMS to detect C4 nitric anhydrides (peroxyisobutyric nitric anhydride, PiBN and peroxy-n-butyric nitric anhydride, PnBN) at $m/z$ 87 following thermal dissociation to a peroxy radical, which reacts with I$^-$ to form C$_4$H$_7$O$_2^-$. As the CI-QMS detects the peroxy-acids at the same $m/z$ as the nitric anhydrides with the same carbon-backbone, we would also expect to detect peroxyisobutyric acid and peroxy-n-butyric acid (Phillips et al., 2013). The CI-QMS was set up to measure PAN during IBAIRN and we therefore regularly added NO to our heated inlet to zero the signal from PAN and thus also PiBN and PnBN. The lack of signal modulation at $m/z$ 87 while adding NO enables us to conclude that the contribution of PiBN and PnBN was insignificant, which is consistent with the low mixing ratios of PAN (the dominant nitric-anhydride at this and most locations) observed during IBAIRN. Mixing ratios of PiBN and PnBN, generally associated with anthropogenically influenced air masses, are expected to be low at this site. Similarly, although differentiation between pyruvic acid and peroxyisobutyric / peroxy-n-butyric acid was not possible with our instrument, we expect the C4 peroxy-acids to be present

at very low concentrations in this pristine environment as their organic backbone is derived from organics of mainly anthropogenic origin (Gaffney et al., 1999; Roberts et al., 2002; Roberts et al., 2003). Similar arguments help us to rule out a large contribution on $m/z$ 87 from butanoic acid, which accompanies anthropogenic activity (e.g. traffic emissions, see Mattila et al. (2018)) and would not acquire continuously high concentrations at this site. Assuming similar sensitivities for butanoic and acetic acid, i.e. 0.62 Hz pptv$^{-1}$ (Eger et al., 2019) butanoic acid mixing ratios sometimes exceeding 2.5 ppbv would be required to account for the entire signal at $m/z$ 87. In the absence of independent measurements of butanoic acid during IBAIRN, we can only conclude that it is unlikely to represent a significant fraction of the CI-QMS signal at $m/z$ 87. While the low resolution of the CI-QMS cannot differentiate between molecules of 87.008 amu with the formula $C_3H_3O_3^-$ (the anion from pyruvic acid) and molecules of 87.045 amu with the formula $C_4H_7O_2^-$ (the anion from PiBN, PnBN or butanoic acid) a second measurement of the exact mass of the anion detected at $m/z$ 87 was provided by an Aerodyne high-resolution long time-of-flight chemical ionisation mass spectrometer (HR-L-TOF-CIMS), equipped with iodide ($I^-$) reagent ions (Lopez-Hilfiker et al., 2014; Riva et al., 2019). This instrument was located about 50 m away from the common inlet and sampled at a height of 1.5 m above the ground. Although neither the instrument nor its inlet transmission was calibrated for pyruvic acid, the signals at $m/z$ 214.921 ($C_3H_4O_3 \cdot I^-$) and $m/z$ 87.008 ($C_3H_3O_3^-$) confirmed the assignment of $m/z$ 87 to a molecule with three of each C- and O-atoms, and thus to pyruvic acid (2-oxo-propionic acid) or an isomer thereof such as 3-oxo propionic acid, $HC(O)CH_2C(O)OH$ (also known as formyl acetic acid or malonaldehydic acid). Figure S3 of the supporting information shows that the dominant contribution to $m/z$ 87 is an ion of formula $C_3H_3O_3^-$, which is a factor of ~ 10 larger than that assigned to $C_4H_7O_2^-$. The HR-L-ToF-CIMS, which was operated under conditions that minimised de-clustering, also identified a signal at $m/z$ 214.921 that could be assigned to $C_3H_4O_3 \cdot I^-$, which was about a factor of 10 higher than for the fragment at $m/z$ 87.008. The correlation coefficient between both signals was 0.77, the deviation from unity likely being related to different response to ambient relative humidity for formation and detection of the cluster and fragment. Pyruvic acid has been detected previously using a HR-L-ToF-CIMS (Lee et al., 2014) whereby a strong dependence of the sensitivity on the relative humidity was observed. If the same factors apply to the instrument used during IBAIRN, a significant change in sensitivity (up to a factor of 2) would have been observed over the course of the diel cycle when RH varied, for example, from 50 % at noon to 100 % at night. One might also expect a reduction in inlet transmission for this soluble, sticky trace gas at high relative humidity. As we have reported previously from the IBAIRN campaign (Liebmann et al., 2018) differences in mixing ratios of trace gases measured using the inlet at 8 m (e.g. CI-QMS) and that at 1.5 m (e.g. HR-L-ToF-CIMS) were great, and especially for soluble trace-gases, displayed different diel profiles due to the impact of ground-level fog in the evenings at the lower level. For these reasons, the uncalibrated HR-L-ToF-CIMS signal is used only to support the identification of pyruvic acid at $m/z$ 87. We cannot rule out that 3-oxo propionic acid contributed to our CI-QMS signal at $m/z$ 87 (or the HR-L-ToF-CIMS signal at $m/z$ 87.008). However, as 3-oxo propionic acid has only been observed in the particle-phase and is associated with air masses impacted by biomass burning (Pillar and Guzman, 2017) our assumption that pyruvic acid is the dominant contributor to the signal at $m/z$ 87 during IBAIRN appears justified.

## 2.2 Other trace gases and meteorological parameters

As well as pyruvic acid, the CI-QMS also measured mixing ratios of PAN, $SO_2$, HCl and a combined signal due to acetic and per-acetic acid. These measurements are described in Eger et al. (2019).

Measurements of $O_3$, NO, $NO_2$, VOCs and meteorological parameters (T, RH, wind speed and direction, photolysis rate coefficients and UVB-radiation) during IBAIRN have recently been described in detail (Liebmann et al., 2018; Liebmann et al., 2019). Briefly, $O_3$ was measured by a commercial ozone monitor (2B-Technology, Model 202) based on optical absorption spectroscopy with a LOD of 3 ppbv (10 s) and a total uncertainty of 2 % $\pm$ 1 ppbv. NO was monitored using a chemiluminescence detector (CLD 790 SR, ECO Physics, Dürnten, Switzerland) with a LOD of 5 pptv (60 s) and a measurement uncertainty of 20 %. The $NO_2$ dataset was provided by a 5-channel, thermal dissociation cavity ring-down spectrometer (TD-CRDS) with a LOD of 60 pptv (60 s) and a total uncertainty of 6 % (Sobanski et al., 2016). CO was measured by a quantum cascade laser (QCL) spectrometer with a total uncertainty of < 20 %. VOC measurements (isoprene and monoterpenes) were performed with a gas chromatograph (Agilent 7890B GC) coupled to an atomic emission detector (JAS AEDIII, Moers, Germany) with an accuracy of 5 % (see supplement of Liebmann et al. (2018)). The GC-AED provides useful information on the contribution of α-pinene, β-pinene, Δ-3-carene, camphene and d-limonene to the sum of monoterpenes. Isoprene and total monoterpenes were additionally measured with a proton transfer reaction time of flight mass spectrometer (PTR-TOF 8000, Ionicon Analytic GmbH) (Jordan et al., 2009; Graus et al., 2010), which was located about 170 m away in dense forest, sampling at a height of 2.5 m above ground. As the PTR-ToF-MS provides a higher temporal resolution than the GC-AED (~ 1 data point per hour), we used this dataset to investigate potential co-variations of pyruvic acid with isoprene and total monoterpenes, bearing in mind that the mixing ratios of monoterpenes observed at the two locations sometimes differ owing to an inhomogeneous distribution of sources and poor mixing within the canopy (Liebmann et al., 2018).

Temperature and relative humidity were measured at the common inlet as well as on the nearby SMEAR II tower at a height of 8 m above ground. Wind direction and speed were measured on the SMEAR II tower (8 m) along with Ultraviolet-B radiation (UVB, 280-320 nm, Solar Light SL501A radiometer, 18 m); the data was provided via SMART-Smear (Junninen et al., 2009). Photolysis rate coefficients ($J_{NO2}$, $J_{NO3}$ and $J_{pyr}$) were calculated from actinic flux measurements at 35 m height using a spectral radiometer (Metcon GmbH) and evaluated cross sections and quantum yields (Burkholder et al., 2015). OH radical concentrations were calculated from the correlation of ground-level OH-measurements with ultraviolet B radiation intensity (UVB, in units of W m$^{-2}$) at the Hyytiälä site (Rohrer and Berresheim, 2006; Petäjä et al., 2009; Hellén et al., 2018). To account for gradients in OH between ground-level and canopy height (Hens et al., 2014), the calculated, ground level OH concentrations (50 % uncertainty) were multiplied by a factor of 2.

## 3 Results and discussion

The IBAIRN campaign was characterised by relatively high day-time temperatures for September and frequent night-time temperature inversions which were accompanied by drastic losses of ozone and an increase in monoterpenes in a very shallow nocturnal boundary layer of ~ 35 m compared to ~ 570 m during daytime (Hellén et al., 2018; Liebmann et al., 2018; Zha et al., 2018; Liebmann et al., 2019). The high variability in the boundary layer height over the course of the diel cycle dictated the diel pattern of many of the trace gases. A time series of pyruvic acid mixing ratios together with isoprene, monoterpenes, $NO_x$, $O_3$ and meteorological parameters is presented in Fig. 1. Pyruvic acid was present at mixing ratios of 17–327 pptv, with a mean value of 96 ± 45 pptv and a median of 97 pptv (based on 1740 data points at 10 min temporal resolution). During two periods of a few hours duration (9[th]-10th Sept), operations from a nearby saw-mill were apparent as elevated monoterpene mixing ratios (Eerdekens et al., 2009; Williams et al., 2011; Hakola et al., 2012). The influence of the saw-mill could be confirmed by examining 48 h back trajectories (HYSPLIT, Draxler and Rolph (2011). These periods are highlighted (grey shading) in Fig. 2 which focuses on a section of the campaign in which the pyruvic acid mixing ratios were rather variable and we compare them with those of isoprene and monoterpenes. There is apparent co-variation of pyruvic acid with isoprene and monoterpenes, the night-time maxima resulting from emissions into the very shallow boundary layer. As we discuss later, the mixing ratios of highly soluble pyruvic acid will be more strongly influenced by deposition or scavenging by aqueous particles than isoprene or monoterpenes so that there is no reason to expect continuously high correlation between these trace-gases as meteorological conditions change.

Owing to its large affinity for surfaces, sharp changes in pyruvic acid mixing ratios (timescales of minutes) will be smeared out over timescales of 10s of minutes because of adsorption and desorption on the inlet line and the filter and filter-holder. We do not expect that this will significantly impact the pyruvic acid time-series over the course of the diel cycle.

No correlation ($R^2 < 0.1$) was found between pyruvic acid mixing ratios and actinic flux, temperature or relative humidity and there was no indication of elevated pyruvic acid mixing ratios in anthropogenically influenced air masses, marked by high levels of $NO_x$. Below we show that known photochemical sources of pyruvic acid are insufficiently strong to account for the observed mixing ratios.

### 3.1 Sources and sinks of pyruvic acid

Figure 3 shows a diel profile of median pyruvic acid, isoprene and monoterpenes, for the whole IBAIRN campaign. Diel mixing ratios of OH, $O_3$ and the rate constant for photolysis of pyruvic acid ($J_{pry}$) are displayed in Fig. 4. The diel profile of pyruvic acid neither follows the actinic flux (or OH) nor $O_3$ (markers of photochemical activity), but has features in common with isoprene and total monoterpenes including a rapid increase between 15 and 17 UTC prior to a decrease in mixing ratio towards midnight. The diel patterns observed are mainly determined by the interplay between production/emission rate (dependent on temperature and light), the boundary layer height and chemical and physical loss processes, such as dry deposition. On nights impacted by strong temperature inversions, the 17:30 maximum was more pronounced indicating the

important role of boundary layer dynamics. The diel profile of pyruvic acid bears more resemblance to that of isoprene than to that of monoterpenes, which may indicate that the emission rate is sensitive to both temperature and levels of photosynthetically active radiation. It is also conceivable that pyruvic acid is not only emitted by the same vegetation as isoprene or monoterpenes but that emissions from undergrowth and decaying vegetation may play a role during the autumn.

Enclosure experiments would be useful to clarify this.

Combining measurement of the mixing ratios of isoprene, monoterpenes and pyruvic acid with calculated loss rates of each enables rough estimation of the source strength of pyruvic acid relative to that of monoterpenes or isoprene. The rate constant for reaction with of OH with pyruvic acid is low (Mellouki and Mu, 2003) so that its main chemical sink during the day is photolysis, with a photolysis rate coefficient of $J_{pyr} \sim 4 \times 10^{-5}$ s$^{-1}$ at solar noon (~ 10:00 UTC, see Fig. 4). The high

solubility of pyruvic acid (see above) implies that dry deposition will be an important sink. To assess its impact on pyruvic acid lifetimes we use the day-time deposition velocity for H$_2$O$_2$ ($V_{dep} = (8 \pm 4)$ cm s$^{-1}$) previously reported for this location (Crowley et al., 2018). The rationale for using the deposition velocity for H$_2$O$_2$ as surrogate for pyruvic acid is a similar solubility ($H_{H2O2} \sim 1 \times 10^5$ Mol atm$^{-1}$). Using $k_{dep} = V_d\ h^{-1}$ and a boundary layer height ($h$) of 570 m at solar noon (Hellén et al., 2018) results in a loss rate constant for deposition of $k_{dep} = 1.6 \times 10^{-4}$ s$^{-1}$. We also consider the loss of pyruvic acid via

heterogeneous uptake to particles, which can be assessed via Eq. (1).

$$k_{het} = \frac{\gamma\ \bar{c}\ A}{4}$$  (1)

where $\gamma$ is the uptake coefficient, $A$ the aerosol surface area density (in cm$^2$ cm$^{-3}$), $\bar{c}$ the average thermal velocity (in cm s$^{-1}$). Using, the mean aerosol surface area observed during IBAIRN of $2 \times 10^{-7}$ cm$^2$ cm$^{-3}$ (Liebmann et al., 2019), with $\bar{c} = 2.65 \times 10^4$ cm s$^{-1}$ at 290 K and an uptake coefficient of 0.06 reported for the uptake to aqueous surfaces (Eugene et al., 2018) we

derive $k_{het} = 8 \times 10^{-5}$ s$^{-1}$.

The overall loss rate (photolysis + deposition  heterogeneous loss) of pyruvic acid is then $L_{pyr} = 2.8 \times 10^{-4}$ s$^{-1}$, corresponding to a lifetime of ~ 1h. We emphasise that the calculated lifetime (and thus the source strength we derive below) are very sensitive to the estimated deposition rate and are thus subject to major uncertainties. In addition, the appropriate uptake coefficient may be less than its value on pure water if the aerosol contains a large mass fraction of organic material which

will reduce the rate of accommodation of pyruvic acid at the surface as has been seen for other trace gases e.g. N$_2$O$_5$ (Folkers et al., 2003; Abbatt et al., 2012).

To calculate the lifetime and the emission rates (at 10:00 UTC) of isoprene and monoterpenes, we assumed that reaction with OH (at $1.5 \times 10^6$ molecule cm$^{-3}$, see Fig. 4) and O$_3$ are the main loss processes and that dry deposition is insignificant. For isoprene, we used the rate coefficients evaluated by IUPAC (Atkinson et al., 2006; IUPAC, 2019), for the monoterpenes we

used the rate coefficients (also from IUPAC) for α-pinene, which constituted, on average, more than 50 % of the overall monoterpene mixing ratio. This resulted in a loss rate constant for isoprene ($L_{isop}$) and monoterpenes ($L_{MT}$) of $1.4 \times 10^{-4}$ s$^{-1}$ for both trace gases, corresponding to a lifetime of ~ 2 h.

Assuming steady state (ss) for all three trace gases, the source strength for pyruvic acid ($S_{pyr}$) relative to the emissions rates of isoprene ($E_{isop}$) or monoterpenes ($E_{MT}$) is given by Eq. (1) and (2) where [pyr], [isop] and [MT] are the measured mixing ratios of pyruvic acid, isoprene and monoterpenes.

$$\frac{S_{pyr}}{E_{isop}} = \frac{[pyr]_{ss}L_{pyr}}{[isop]_{ss}L_{isop}} \tag{2}$$

$$\frac{S_{pyr}}{E_{MT}} = \frac{[pyr]_{ss}L_{pyr}}{[MT]_{ss}L_{MT}} \tag{3}$$

Taking the diel averaged mixing ratios of pyruvic acid, isoprene and monoterpenes at 10:00 UTC (83, 22 and 168 pptv) and the loss terms calculated above, results in a pyruvic acid source strength relative to isoprene and monoterpenes (based on the PTR-ToF-MS measurements) of 7 and 0.8, respectively (Table 1). When using the (low time resolution) GC-AED dataset for isoprene and monoterpenes, these values increase to 14 and 1.3, respectively.

In steady-state, using $S_{pyr} = [pyr]_{ss}L_{pyr}$, the pyruvic acid source strength needed to account for the observed 10:00 UTC mixing ratios of ~ 80 pptv is $S_{pyr}$ = 80 pptv h$^{-1}$ (or 12 pptv h$^{-1}$ when neglecting dry deposition and heterogeneous uptake to particles, see Table 1). These values can be compared with the production rate of pyruvic acid from the photochemical oxidation of isoprene, which we calculate to be 0.02 pptv h$^{-1}$, orders of magnitude too small to explain the pyruvic acid mixing ratios observed. The basis for this calculation were measured isoprene and O$_3$ mixing ratios and the results from chamber experiments (Grosjean et al., 1993; Paulot et al., 2009) that report a pyruvic acid yield from isoprene oxidation of ~ 2 %. Pyruvic acid is also a product of the ozonolysis of methyl vinyl ketone (MVK), with a yield of ~ 5 % (Grosjean et al., 1993). In order to explain the observed pyruvic acid mixing ratios by the production rate from MVK alone would require 16 ppbv of MVK, which is a factor ~ 60 more than observed at this site during Sept. (Hakola et al., 2003) and clearly not feasible.

As the degradation of monoterpenes is not expected to generate pyruvic acid (Vereecken et al., 2007; IUPAC, 2019), we conclude that, in the boreal forest around Hyytiälä, the main source of pyruvic acid is direct emission from the biosphere and not photochemical production via reactions of OH or O$_3$. This is consistent with the measurements of Jardine et al. (2010) who report high mixing ratios of pyruvic acid resulting from direct emission in an O$_3$ and OH-free environment. In contrast, Mattila et al (2018) provide convincing evidence for photochemical production of pyruvic acid resulting in a mean mixing-ratio of 180 pptv maximising with photochemical activity. Mattila et al. (2018) also observed a strong reduction in the mixing ratio of pyruvic acid with height within the boundary layer due to dry deposition and found no evidence for strong surface emissions. As described above, dry deposition will also have impacted on the pyruvic acid mixing ratios observed at Hyytiälä, though in the absence of vertical profiles or flux measurements it is difficult to assess rigorously its impact during day or night. The differences between the summertime measurements of Mattila et al. (2018) and the present work are very likely related to the starkly contrasting environments: The IBAIRN campaign being conducted in the remote, boreal forest in Autumn whereas Mattila et al (2018) made their summertime measurements over an eight-day period in a region with sparse vegetation and with significant anthropogenic influence from traffic, oil and natural gas operations and livestock and NO$_X$

levels of several ppbv, and concluded that pyruvic acid was generated photochemically from aromatics emitted by automobiles.

## 3.2 Role of gas-phase pyruvic acid in the troposphere

In this section, we assess the potential role of pyruvic acid as source of radicals and other reactive trace gases in the boreal forest. Figure 5 provides an overview of the sources and sinks of pyruvic acid. The dominant photochemical loss process of pyruvic acid is its photolysis. Experimental data on its UV-cross-sections and photodissociation quantum yields have recently been evaluated by the IUPAC panel (IUPAC, 2019). The thermodynamically accessible dissociation pathways are listed below (R1–R3).

$CH_3C(O)C(O)OH + h\nu$ $\rightarrow$ $CH_3CHO + CO_2$ (R1)

$\rightarrow$ $CH_3CO + HOCO$ (R2)

$\rightarrow$ $CH_3C(O)OH + CO$ (R3)

Photolysis of gas-phase pyruvic acid in the actinic region ($\lambda > 310$ nm) results mainly in the formation of acetaldehyde $CH_3CHO + CO_2$ (R1) with a yield of 60 %. The second most important channel (R2, with a yield of 35 %) leads to formation of organic radical fragments which react with $O_2$ to form the peroxy radicals $CH_3C(O)O_2$ and $HO_2$ (reactions R4 and R5).

$CH_3CO + O_2 + M$ $\rightarrow$ $CH_3C(O)O_2 + M$ (R4)

$HOCO + O_2$ $\rightarrow$ $HO_2 + CO_2$ (R5)

Acetaldehyde (formed in R1) is an air pollutant which plays an important role in tropospheric chemistry as a source of PAN (Roberts, 1990), PAA (Phillips et al., 2013; Crowley et al., 2018), $HO_x$ radicals (Singh et al., 1995) and ultimately, via methyl peroxy radical oxidation, HCHO (Lightfoot et al., 1992). Based on campaign-median pyruvic acid mixing ratios and photolysis rates measured during IBAIRN (see Figs. 3 and 4), we calculate an acetaldehyde production rate of $P_{CH3CHO} = 0.6$ $J_{pyr} \times [pyr] = 6.3$ pptv h$^{-1}$ at local noon (Table 2, Fig. S4).

On a global scale the main source of acetaldehyde is OH-initiated photochemical production from alkanes, alkenes, ethanol and isoprene with alkanes accounting for about one half of the total production of 128 Tg a$^{-1}$ (Millet et al., 2010). Minor sources are direct biogenic emissions (23 Tg a$^{-1}$), anthropogenic emissions (2 Tg a$^{-1}$) and biomass burning (3 Tg a$^{-1}$). As alkanes were not measured during IBAIRN we estimate the mixing ratios of the three most abundant alkanes (ethane, propane and n-butane) from monthly averages measured at Pallas and Utö (both Finland) for the years 1994-2003 (Hakola et al., 2006), which are consistent with measurements at Pallas in 2012 reported by Hellén et al. (2015). Combining mean (September) mixing ratios of 1000 pptv of ethane, 250 pptv of propane and 150 pptv of n-butane with OH rate coefficients of $k_{OH+ethane} = 2.4 \times 10^{-13}$ cm$^3$ molecule$^{-1}$ s$^{-1}$, $k_{OH+propane} = 1.1 \times 10^{-12}$ cm$^3$ molecule$^{-1}$ s$^{-1}$ and $k_{OH+n-butane} = 2.35 \times 10^{-12}$ cm$^3$ molecule$^{-1}$ s$^{-1}$ at 298 K (IUPAC, 2019) and acetaldehyde yields (at 0.1 ppbv of $NO_x$) of 0.50, 0.24 and 0.69 (Millet et al., 2010), results in a total $CH_3CHO$ production rate from OH + alkanes of 2.2 pptv h$^{-1}$ at local noon (Table 2). Figure S4 indicates how these production rates vary over the diel cycle. From these calculations we conclude that emission and

subsequent photolysis of pyruvic acid is likely an important source of $CH_3CHO$ in this environment and may impact our current understanding of the acetaldehyde budget (Millet et al., 2010) in forested regions in general.

The dominant sink of $CH_3CHO$ is the reaction with OH (R6) with a rate constant of $1.5 \times 10^{-11}$ cm$^3$ molecule$^{-1}$ s$^{-1}$ (IUPAC, 2019) to form the $CH_3CO$ radical. This then reacts in air (R4) to form $CH_3C(O)O_2$ which is the precursor of PAN ($CH_3C(O)O_2NO_2$, R7), peracetic acid ($CH_3C(O)OOH$, R8), acetic acid ($CH_3C(O)OH$, R9) and $CH_3O_2$ (R10 and R11) and which can recycle OH (R10).

| | | | |
|---|---|---|---|
| OH + $CH_3CHO$ (+$O_2$) | $\rightarrow$ | $CH_3C(O)O_2$ + $H_2O$ | (R6) |
| $CH_3C(O)O_2$ + $NO_2$ + M | $\rightarrow$ | $CH_3C(O)O_2NO_2$ + M | (R7) |
| $CH_3C(O)O_2$ + $HO_2$ | $\rightarrow$ | $CH_3C(O)OOH$ + $O_2$ | (R8) |
| | $\rightarrow$ | $CH_3C(O)OH$ + $O_3$ | (R9) |
| | $\rightarrow$ | OH + $CH_3O_2$ + $CO_2$ | (R10) |
| $CH_3C(O)O_2$ + NO ($O_2$) | $\rightarrow$ | $CH_3O_2$ + $CO_2$ + NO$_2$ | (R11) |

The second most important photolysis channel for pyruvic acid is Reaction (R2), which leads to formation of $HO_2$ and $CH_3C(O)O_2$. These radicals play a crucial role in photochemical ozone production (Fishman and Crutzen, 1978), in the recycling of OH (in the presence and absence of $NO_x$) and, as described above, in PAN formation.

The production rate (10:00 UTC) of $HO_2$ and $CH_3C(O)O_2$ from pyruvic acid photolysis is given by $P_{HO2} = P_{CH3CO3} = 0.35$ $J_{pyr} \times$ [pyr] and is equal to 4 pptv h$^{-1}$. This value is roughly equivalent to the production rate of $CH_3C(O)O_2$ from the OH-initiated acetaldehyde oxidation (the major source of this radical) assuming typical values of 100 pptv $CH_3CHO$ and $1.5 \times 10^6$ OH molecule cm$^{-3}$ and using $k_{OH+CH3CHO} = 1.5 \times 10^{-11}$ cm$^3$ molecule$^{-1}$ s$^{-1}$ (IUPAC, 2019). We therefore conclude that pyruvic acid photolysis in this environment is an important source of the $CH_3C(O)O_2$ radical both directly (R2 and R4) and via acetaldehyde formation (R1 and R6).

Taking median $O_3$ and CO mixing ratios (at 10:00 UTC) found in IBAIRN, we can easily show that the rate of $HO_2$ formation (4 pptv h$^{-1}$) from pyruvic acid photolysis (reactions R2 and R5) is minor compared to that from OH + $O_3$ of 12 pptv h$^{-1}$ (with $k_{OH+O3} = 7.3 \times 10^{-14}$ cm$^3$ molecule$^{-1}$ s$^{-1}$) and OH + CO of 100 pptv h$^{-1}$ (with $k_{OH+CO} = 2.1 \times 10^{-13}$ cm$^3$ molecule$^{-1}$ s$^{-1}$) (IUPAC, 2019). It is also negligible compared to the total $HO_2$ production rate of 100–600 pptv h$^{-1}$ previously derived in this environment (albeit in summer) via measurement of $HO_x$ (Hens et al., 2014).

So far, to calculate the photo-dissociation rate constant for pyruvic acid ($J_{pyr}$) we have used the IUPAC recommendation of an overall quantum yield of 0.2 at atmospheric pressure. There are however several inconsistencies in the experimental data sets on pyruvic acid photolysis, with two groups reporting quantum yields that are a factor of ~ 4 larger at this pressure (Berges and Warneck, 1992; Reed Harris et al., 2017). If these large quantum yields were to be correct, the calculated production rates of $CH_3CHO$ and $CH_3C(O)O_2$ would increase by a factor of 4 (see Table 2) so that $P_{CH3CHO} = 28$ pptv h$^{-1}$ (Table 2). Moreover, Reed Harris et al. (2017) report much lower yields of $CH_3CHO$, and suggest that other processes may compete with rearrangement of the methyl-hydroxycarbene ($CH_3COH$) necessary to form acetaldehyde. They propose that in

air, initially formed methyl-hydroxycarbene may react with $O_2$ to form $CH_3CO$ and $HO_2$. If this is correct, the intermediate step (R6) in which OH reacts with acetaldehyde to form $CH_3C(O)O_2$ in air, is bypassed, so that pyruvic acid photolysis would be an even more important source of PAN. This alternative fate of the methyl-hydroxycarbene radical is depicted with the dashed line in Fig. 5.

## 4 Conclusions

Mixing ratios of pyruvic acid of 17–327 pptv (mean of $96 \pm 45$ pptv) were measured in the boreal forest in Hyytiälä, southern Finland, during a field study in late summer (September 2016). Campaign averaged, diel profiles of pyruvic acid displayed similar features to those of monoterpenes and isoprene. Combining the mixing ratios of pyruvic acid with its loss terms enabled calculation of the source strength at solar noon of $\sim 60$ pptv h$^{-1}$. There appears to be no known photochemical mechanism to generate pyruvic acid at this rate and we suggest that pyruvic acid is, to a large extent, emitted directly from the biosphere. We show that pyruvic acid, at the mixing ratios observed in September, represents an important source of acetaldehyde and the acetyl peroxy radical, thus enhancing the formation of PAN, $C_2$-organic acids and $CH_3O_2$.

We conclude that, during late summer / autumn, pyruvic acid is an important biogenic VOC in the boreal forest which has previously received little attention. Further field and enclosure studies are necessary to quantify its emissions and role during other seasons and to better understand its sources and sinks (e.g. generation in $OH/O_3/NO_3$ initiated oxidation of terpenes and dry deposition rates) in the boreal forest as well as in other environments. To this end, co-located, high-time-resolution measurements of mixing ratios and fluxes of terpenoids and pyruvic acid are necessary.

In addition, further laboratory studies are required to resolve discrepancies in the literature data on the pressure (and wavelength) dependence of both the overall photolysis quantum yield and the product distribution during pyruvic acid photolysis in the gas-phase.

## Data availability

The data used in this study are archived with Zenodo at https://doi.org/10.5281/zenodo.3374518. Depending on agreement with the IBAIRN data protocol, the data will be available for external users from August 2019.

## Author contributions

PE performed the CI-QMS measurements of pyruvic acid during IBAIRN, analysed the data and, with contributions from JC and JL, wrote the manuscript. NS was responsible for the CRDS measurements of $NO_2$. JS was responsible for the $O_3$ and photolysis rate coefficient measurements. HF was responsible for the NO and CO measurements. MR and QZ were responsible for the HR-L-ToF-CIMS measurements of pyruvic acid. EK and JW provided the GC-AED measurements of

isoprene and individual monoterpenes. LQ and SS contributed the PTR-ToF-MS measurements of isoprene and monoterpenes. The IBAIRN campaign was conceived and organised by JC and ME. All authors contributed to the manuscript.

## Competing interests

5 The authors declare that they have no conflict of interest.

## Acknowledgements

We would like to thank Uwe Parchatka for the provision of NO measurements and Janne Levula and the team at Hyytiälä for the excellent technical support.

## Financial support

10 This work was supported by ENVRIplus, the European Research Council (grant 638703-COALA) and the Academy of Finland Centre of Excellence program (project numbers 307331, 317380 and 320094).

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

**Table 1:** Calculated source strength of pyruvic acid ($S_{pyr}$), production rate from isoprene + $O_3$ ($P_{pyr\ (isop+O3)}$) and emission rates of isoprene ($E_{isop}$) and monoterpenes ($E_{MT}$) at solar noon.

| Source strength [pptv h$^{-1}$] / emission rate | $J_{pyr} = 4 \times 10^{-5}$ s$^{-1}$ $k_{dep} = 16 \times 10^{-5}$ s$^{-1}$ | $J_{pyr} = 4 \times 10^{-5}$ s$^{-1}$ $k_{dep} = 0$ |
|---|---|---|
| $S_{pyr}$ | 60 | 12 |
| $P_{pyr\ (isop+O3)}$ | 0.02 | |
| | PTR-ToF-MS | GC-AED |
| $E_{isop}$ | 12 | 6 |
| $E_{MT}$ | 94 | 65 |

Notes: $S_{pyr}$ is the net source strength (emission rate + production rate) of pyruvic acid based on measured mixing ratios at solar noon, the assumption of steady-state and that photolysis ($J_{pyr}$) and deposition ($k_{dep}$) are the only significant loss processes. The net source strength is derived for two different values of $k_{dep}$ as discussed in the text. $P_{pyr\ (isop+O3)}$ is the rate of photochemical production of pyruvic acid from isoprene oxidation. The net isoprene and net monoterpene emission rates ($E_{isop}$ and $E_{MT}$) were calculated using their mixing ratios and considering the reactions with OH and $O_3$ as the only relevant loss terms. Emission rates are shown for both VOC datasets (PTR-ToF-MS and GC-AED).

**Table 2:** Calculated production rates of acetaldehyde, $HO_2$ and $CH_3C(O)O_2$ from the photolysis of pyruvic acid at solar noon.

| $CH_3CHO$ production rate [pptv h$^{-1}$] | $J_{pyr} = 4 \times 10^{-5}$ s$^{-1}$ | $J_{pyr} = 16 \times 10^{-5}$ s$^{-1}$ |
|---|---|---|
| pyruvic acid + h$\nu$ | 6.3 | 25.2 |
| OH + ethane | 0.6 | 0.6 |
| OH + propane | 0.3 | 0.3 |
| OH + n-butane | 1.3 | 1.3 |
| $HO_2$ production rate [pptv h$^{-1}$] | | |
| pyruvic acid + hv | 4 | 16 |
| OH + O$_3$ | 12 | 12 |
| OH + CO | 100 | 100 |
| $CH_3C(O)O_2$ production rate [pptv h$^{-1}$] | | |
| Pyruvic acid + h$\nu$ | 4 | 16 |
| $CH_3CHO$ + h$\nu$ | 5 | 5 |

Notes: The production rates of $CH_3CHO$, $HO_2$ and $CH_3C(O)O_2$ from pyruvic acid photolysis are derived for two different values of $J_{pyr}$ using quantum yields of 0.2 and 0.8 (see text). The production rates of $CH_3CHO$ formation from alkanes are based on estimated mixing ratios (see text). The production rate of $HO_2$ from the reaction of OH with $O_3$ and CO is based on calculated OH and measurements of $O_3$ and CO during IBAIRN. The production rate of $CH_3C(O)O_2$ from $CH_3CHO$ was calculated using a mixing ratio of 100 pptv of acetaldehyde.

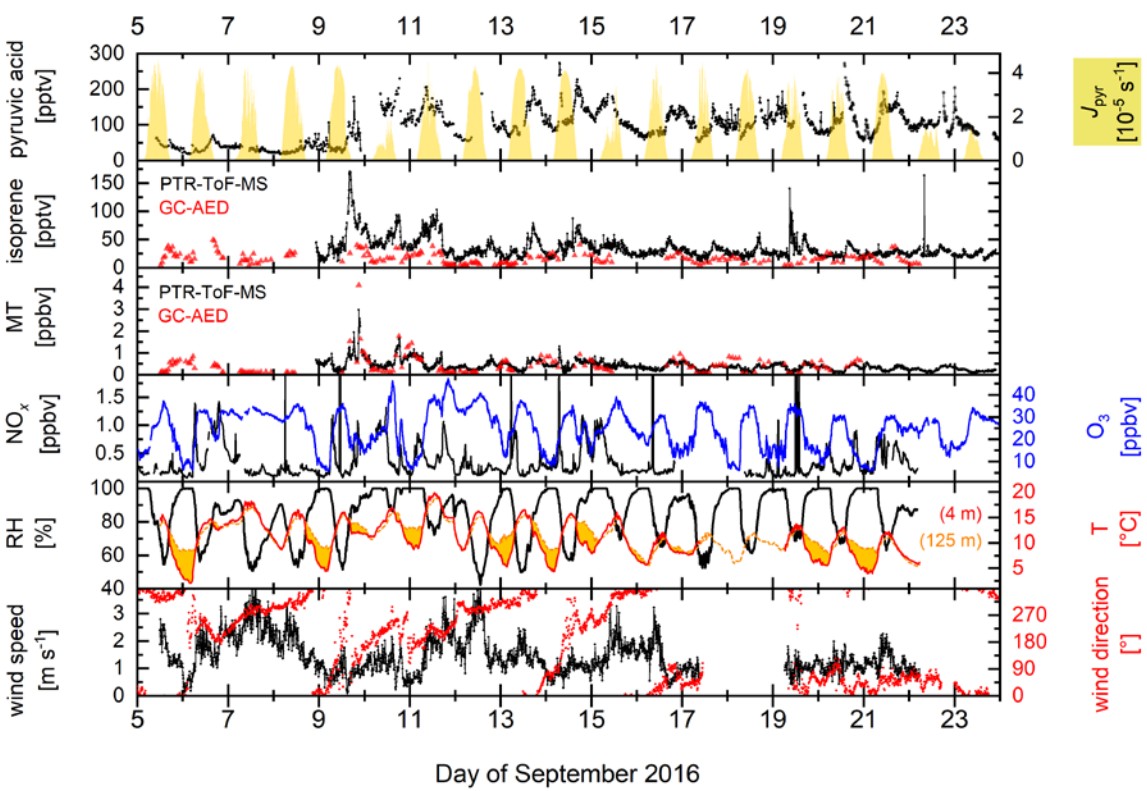

**Figure 1:** Time series of pyruvic acid mixing ratios, photolysis rate ($J_{pyr}$), isoprene and monoterpenes (PTR-ToF-MS, GC-AED), $NO_x$ ($NO_2$ + NO), $O_3$, RH, temperature (at 4 and 125 m, with night-time inversions indicated by the coloured area) and wind speed and direction during the IBAIRN campaign.

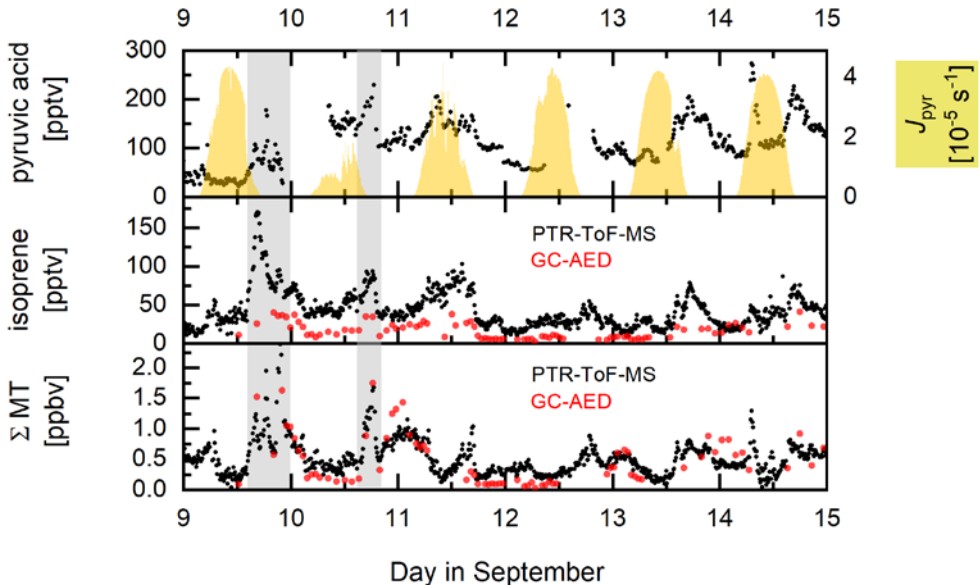

**Figure 2:** Times series of pyruvic acid, isoprene and total monoterpenes (Σ MT). The shaded areas represent periods where the site was impacted by saw-mill activity.

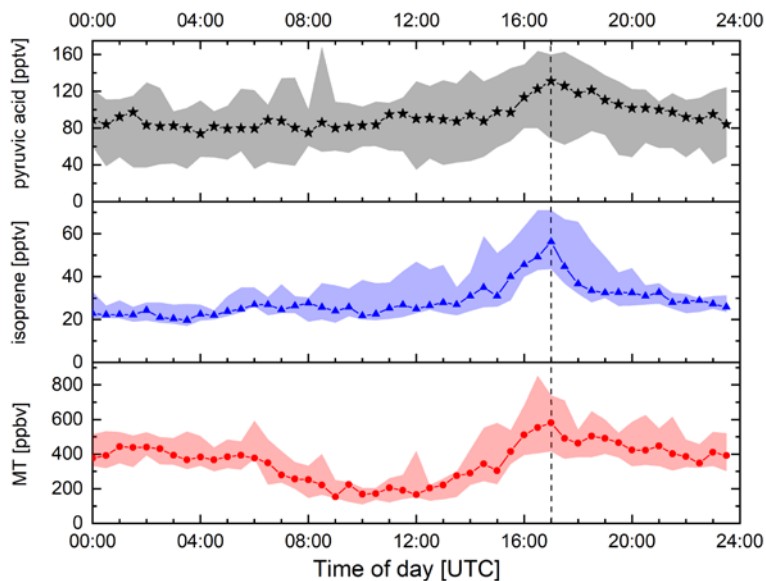

**Figure 3:** Diel profiles of median mixing ratios of (a) pyruvic acid, isoprene and monoterpenes (MT) during the IBAIRN campaign. The shaded area represents the 25th and 75th percentiles.

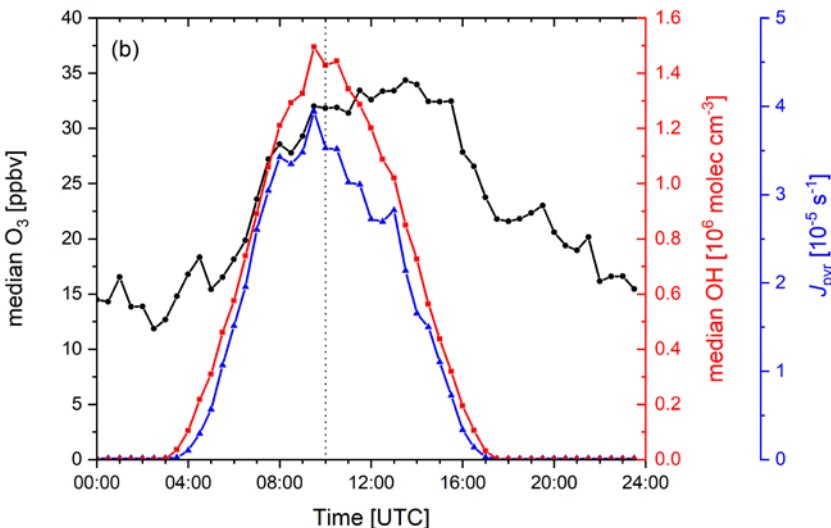

**Figure 4:** Diel profiles of median mixing ratios of $O_3$, OH and $J_{pyr}$ (calculated with a quantum yield of 0.2) during the IBAIRN campaign.

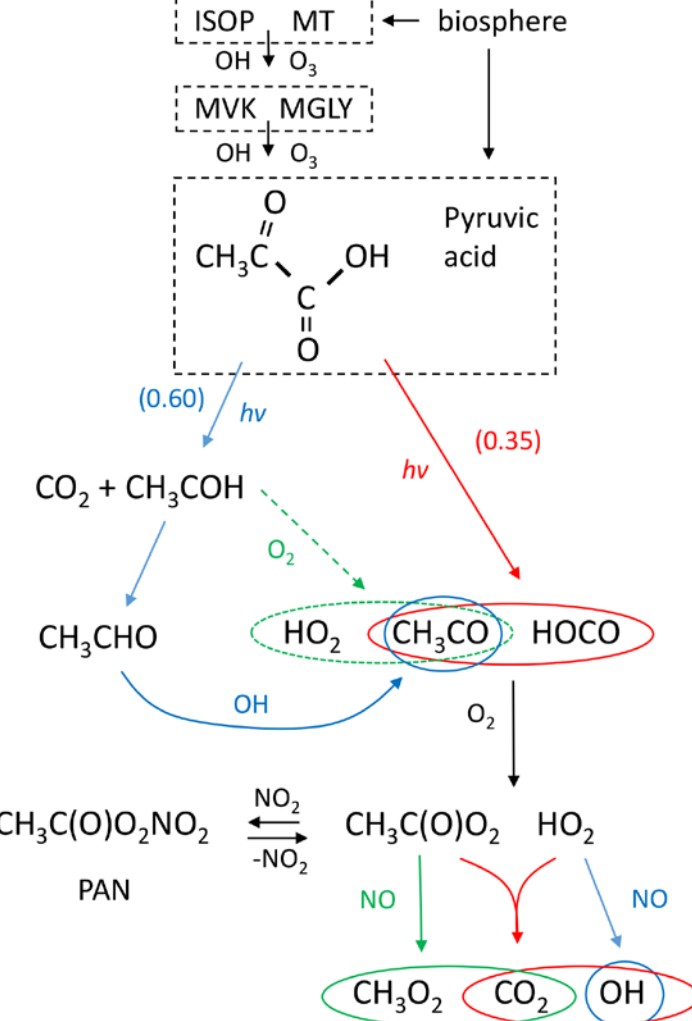

**Figure 5:** Sources of pyruvic acid and mechanism of formation of CH3CHO, HO2 and CH3C(O)O2 during its photolysis. ISOP = isoprene, MT = monoterpenes, MVK = methyl vinyl ketone, MGLY = methylglyoxal. Numbers in parentheses indicate branching ratios. CH3COH is the methyl-hydroxycarbene that is believed to mainly rearrange to form CH3CHO.