# Peer review of "Pyruvic acid in the boreal forest: Gas-phase mixing ratios and impact on radical chemistry"

_Atmospheric Chemistry and Physics, 2019_

## Referee Comment (RC1) · Anonymous Referee #1 · 4 Nov 2019

General Comments:

The manuscript presents gas-phase measurements of pyruvic acid in the boreal forest during the IBAIRN campaign. The average mixing ratio of pyruvic acid was 96 pptv with a maximum of 327 pptv. The contribution of isoprene oxidation and direct emission of gaseous pyruvic acid to the source strength is assessed in this work. The photolytic fate of gaseous pyruvic acid results in the production of acetaldehyde and peroxy radicals in the boreal forest. This is an interesting manuscript for the readers of Atmospheric Chemistry and Physics and should be published after a revision. Specific comments are provided below to improve the manuscript before publication.

Specific Comments:

[Figure]

1) The revised version of the manuscript should change its title by deleting the word "First". Such attribution is unnecessary and conflicts with other previous field measurements of pyruvic acid. In addition, while measuring pyruvic acid as described, the manuscript should indicate what other species were monitored. It is unlikely that the work was specific for pyruvic acid but that a mass range was analyzed. This range should be provided in the text and considerations for correlated species should be made in the revision to avoid fractionation of the data by reporting it together.

2) References to original papers introducing a concept or finding as well as the most recent literature should be both carefully highlighted in the manuscript. For example, in p. 1 l. 28, by reading Magel et al. (2006) it is clear the concept for the wide availability of pyruvic acid in the central metabolism of plants is not introduced by them but actually by Eisenreich et al., Trends in Plant Sciences 2001, 7, 78-84, DOI: 10.1016/S1360-1385(00)01812-4. Similarly, the source for Jardine et al. (2010) could be checked. In addition, in p. 2 l. 13 and l. 15, there have been four recent reports for the heterogeneous photooxidation of aromatics emitted during biomass burning that produced oxocarboxylic acids, including pyruvic acid by Pillar et al., which should be briefly described here: i) Pillar et al., Environmental Science & Technology 2017, 51, 951-4959, DOI: 10.1021/acs.est.7b00232; ii) Pillar et al., Environmental Science & Technology 2014, 48, 14352-14360. DOI: 10.1021/es504094x; iii) Pillar-Little et al., Environments 2018, 5, 104. DOI: 10.3390/environments5090104; and iv) Pillar et al., Journal of Physical Chemistry A 2015, 41,10349-10359. DOI: 10.1021/acs.jpca.5b07914.

3) P. 2 l. 21-23 and p. 8 l. 25: There are some differences with more current literature than the cited, which need to be considered here and estimated during a comparison of phase processes and drivers for pyruvic acid loss in the conclusions sections of the manuscript. Particularly, the photochemical loss of pyruvic acid in water was evaluated by Eugene and Guzman, J. Phys. Chem. A 2017, 121, 15, 2924-2935, DOI: 10.1021/acs.jpca.6b11916, to occur with a lifetime of only ∼22 min, and was indicated to vary by ∼2-times in the gas phase. Similarly, the lifetime for the loss of pyruvic acid

reacting with OH radical in water was evaluated to be 1-10 days and in the gas phase to be 2-months.

4) P. 2 l. 28-30: It would be important to indicate here that one of the factors that enhances the solubility of pyruvic acid, as indicated to be "highly soluble" was explained in a paper by Guzman et al., J. Am. Chem. Soc. 2006, 128, 10621-10624, DOI: 10.1021/ja062039v; in terms of a cooperative reversible equilibrium of hydration that converts the carbonyl group into a geminal diol. Similarly, for the mention of pyruvic acid "partitioning into the aerosol-phase" there should be a description here of recent work discussing the crystal structure of pyruvic acid by Heger et al., Acta Crystalographica E 2019, 75 (6), 858-862, DOI: 10.1107/S2056989019007072, and the novel study of the partitioning of gas phase pyruvic acid into water by Eugene et al., Langmuir 2018, 34, 9307-9313, DOI: 10.1021/acs.langmuir.8b01606. The last Langmuir paper should also be connected to the text about water clusters of pyruvic acid and pyruvate in p. 4 l. 26, and to the text in p. 8 l. 19-21 by including the reactive uptake coefficient of gaseous pyruvic acid by water reported to be 0.06. Finally, connected to the contribution of pyruvic acid to "SOA formation",it would be appropriate to include original related literature and the most updated papers: i) Guzman et al., J. Phys. Chem. A 2006, 110, 3619-3626, DOI: 10.1021/jp056097z; ii) Guzman et al., J. Phys. Chem. A 2006, 110, 931-935, DOI: 10.1021/jp053449t; iii) Xia et al., J. Phys. Chem. A 2018, 122, 6457-6466, DOI: 10.1021/acs.jpca.8b05724, iv) Eugene and Guzman, Molecules 2019, 24, 1124; DOI: 10.3390/molecules24061124 et al., Environ. Sci. Technol. 2019, DOI: 10.1021/acs.est.9b03742; v) Mekic at al., Environ. Sci. Technol. 2018, 52, 12306-12315, DOI: 10.1021/acs.est.8b03196; and vi) Mekic et al., Atmos. Environ. 2019, 219, 117046, DOI: 10.1016/j.atmosenv.2019.117046.

5) P. 4 l. 15: In standard analytical chemistry work the limit of detection (LOD) is established with a 99% chance of being greater than a blank for three times the standard deviation. It is confusing here that a value of 15 pptv is provided with 2 sigma for 10 s. This issue requires clarification.

6) P. 4 l. 23-24: More details about how the oxidation was accomplished in this work (and not in the reference cited) should be provided here. An associated calibration curve should be added to the supplementary information.

7) P. 5 l. 4-5: This statement is unclear. What is the meaning of a 30 % $\pm$ 10 pptv mixing ratio? Clarify the units.

8) From p. 5 l. 22 to p. 6 l. 14, and Figure S1: The general idea and the analysis are acceptable but do not explain that the anion at m/z 87 could also be assigned to 3-oxopropanoic acid.

9) P. 7 l. 21-30: This observation for the activity associated to the saw mill appears anecdotic in the context of evaluating pristine emissions from the natural boreal forest environment. The revisions should consider if the text is contradicting the objective of the manuscript.

10) After the text is revised, and the calculations are rechecked based on the comments above and the inclusion of the reactive uptake coefficient of gaseous pyruvic acid by water in the model, some editions of section 3.2 will be needed in the revised manuscript.

Technical Corrections:

P. 3. l. 20: Delete "but unexplored atmospheric component".

P. 4 l. 18-20: The statement "Laboratory . . . lower" is unnecessary and can be deleted.

P. 9 l. 26: Add a mention to "Gas Phase".

P. 10 l. 3: Insert "gaseous" before pyruvic.

P. 12 l. 9: add "in the gas phase" at the end.

---

## Referee Comment (RC2) · Anonymous Referee #2 · 13 Nov 2019

This paper presents the first measurements of pyruvic acid in a boreal forest. There are pyruvic acid measurements in other locations but not many. In addition, pyruvic acid (unlike many other organic acids) has an active photochemistry and photolyzes rapidly. This allows it serve as a source of acetaldehyde which is probably the most important point in the paper. I think the results are interesting and probably should be published but I think the authors need to address some issues first. 1) I have some concerns about the measurement method. The largest is the use of an inlet that is heated to 200 oC. This is done so the instrument can measure PANs (it would be nice to see these concentrations in the SI at least as well and may help rule out contributions to PiBN which could have a very large sensitivity). However, it may also lead to unwanted chemistry in the heated inlet. Do the authors know that the signal for

pyruvic is present in the CIMS if the sampled air is unheated? Is the measured pyruvic sample modified by heating of the inlet? I assume the aerosol filter before the CIMS takes out all the organic aerosol that might evaporate. However, I would think it might significantly impact the transmission and time response to pyruvic. Was this tested? Could this filter significantly smear out the pyruvic observations? I would be especially concerned at high RH. 2) My second concern about the method is the measurement of the pyruvic background. In addition, I don't understand how the LOD for the instrument is defined. I assume it depends on the variability of the background? So this should be clearly defined. I also don't understand how the background is measured. It seems like the steel wool scrubber doesn't react fast and so the authors chose not to use this as a background measurement? It is also stated that the background varies with the overall signal. OK if this is true then at least show this in the SI. There is also a mention that backgrounds are derived when the measurement is near LOD or when the instrument is overflowed with synthetic air. So I really don't understand how the background signal is measured and interpolated as function of time. I think this needs to be clearly explained in detail with graphical examples at least given in the SI. I also don't understand how overflowing with zero air is considered preferable to scrubbing ambient air – I am sure it is easier to deal with but I am not sure it is valid. Especially if the zero air is a different RH or temperature than ambient. My guess is that pyruvic acid is similar to nitric acid and the whole key to the measurement may be accurately measuring the instrument background. 3) Did you measure formic or acetic acid during the campaign? Formic should be readily apparent at its cluster with I-. Is there evidence for acetic acid at 59? The presence of butanoic acid is ruled out in part by assuming a similar sensitivity as acetic acid (i.e. lower than for pyruvic). Why not just measure the butanoic sensitivity? If there are significant levels or formic and acetic then you might expect other carboxylic acids as well. I also don't think you can argue that the lack of butanoic in the Matilla et al work argues that it is not present in this region. The pyruvic acid is certainly very different in many respects between the two locations if this is true. 4) The sensitivity is reported per MHz of I-. What was the typical sensitivity? In addition,

if I(CN)2- is the true reagent ion why report sensitivity relative to I-? How were post mission calibrations related to the field data? 5) The Jardine et al., 2010 observations were done in a biosphere and are not truly ambient and in my opinion are closer to a chamber experiment. So I would not classify them as ambient measurements. 6) I don't understand including the contaminated "sawmill" days in the data. I think it only confuses things. I would remove from the dataset and concentrate on the clean boreal forest conditions. 7) I am not sure I agree with the following statement "On several days the pyruvic acid mixing ratios co-vary with those of isoprene and monoterpenes, with night-time maxima resulting from emissions into the very shallow boundary layer, which is especially apparent in the period 9–15 20 September 2016". I certainly can't make it out from the way the data is presented. If this statement is to be maintained it needs to be backed up with a figure that shows it more clearly. The use of the GC vs. PTR-MS data for isoprene and monoterpenes also needs to be clarified as they look pretty different. Is there a measurement of boundary layer height to back up this statement? If it co-varies some days why not others? Also I think Figure 3 is misleading. I am not convinced that you have a regular diurnal variation in pyruvic. There needs to be error bars in this figure that represents the deviation from average. 8) WRT the pyruvic observations I am most bothered/intrigued that it doesn't seem to go to zero at night for the latter part of the campaign. This is very different than the Matilla et al. work which shows that the pyruvic goes to zero at night and also is greatly diminished during the day at lower altitudes. This implies efficient dry deposition in Colorado that I think should be active in Finland as well. Given that the RH goes to 100% on several nights as well I would expect dew formation and even greater enhancements in the loss of pyruvic at night. This needs to be addressed by the authors. Do they think the loss of pyruvic to the forest at night is minimized? Do emissions need to go up at night to explain observations? 9) I am also struck by the lack of correlation of pyruvic with temperature, J, etc. If pyruvic is truly emitted by vegetation then I think at the least it should be related fairly strongly to temperature. Is there an example of plant metabolite emissions that are unrelated to temperature? or PAR? 10) In summary,

I think the authors have demonstrated that the observed pyruvic levels are not due to isoprene or terpene photochemistry. However, I am not totally convinced they are observing only pyruvic acid. I am also skeptical that vegetation would emit pyruvic in a manner needed to explain the observations. So I do think that the authors need to restate their conclusions (i.e. in abstract) especially since they have not demonstrated a flux of pyruvic to the atmosphere. I agree that further measurements including flux and altitude profiles would be very useful to sort this out. It would be very interesting if the boreal forest emits as much pyruvic acids as monoterpenes to the atmosphere.

---

## Author Comment (AC1) · 9 Dec 2019

**Reply to RC1**

*In the following, the referee's comments are reproduced (black) along with our replies (blue) and changes made to the text (red) in the revised manuscript.*

**General statement:**

The manuscript presents gas-phase measurements of pyruvic acid in the boreal forest during the IBAIRN campaign. The average mixing ratio of pyruvic acid was 96 pptv with a maximum of 327 pptv. The contribution of isoprene oxidation and direct emission of gaseous pyruvic acid to the source strength is assessed in this work. The photolytic fate of gaseous pyruvic acid results in the production of acetaldehyde and peroxy radicals in the boreal forest. This is an interesting manuscript for the readers of Atmospheric Chemistry and Physics and should be published after a revision. Specific comments are provided below to improve the manuscript before publication.

We thank the referee for the positive evaluation of our manuscript and the useful comments and suggestions. We have modified the manuscript according to the comments listed below.

**Specific comments:**

1) The revised version of the manuscript should change its title by deleting the word "First". Such attribution is unnecessary and conflicts with other previous field measurements of pyruvic acid.

We changed the title by replacing the word "first" with "gas-phase"

"Pyruvic acid in the boreal forest: gas-phase mixing ratios and impact on radical chemistry"

In addition, while measuring pyruvic acid as described, the manuscript should indicate what other species were monitored. It is unlikely that the work was specific for pyruvic acid but that a mass range was analyzed. This range should be provided in the text and considerations for correlated species should be made in the revision to avoid fractionation of the data by reporting it together.

We added a paragraph to the method section describing that we also measured PAN, $SO_2$, HCl and a combined signal of PAA and acetic acid. Further we clarify that this data set has already been published in Eger et al. (2019) which describes the instrument in detail.

As well as pyruvic acid, the CI-QMS also measured mixing ratios of PAN, $SO_2$, HCl and a combined signal due to acetic and per-acetic acid. These measurements are described in Eger et al. (2019).

2) References to original papers introducing a concept or finding as well as the most recent literature should be both carefully highlighted in the manuscript. For example, in p. 1 l. 28, by reading Magel et al. (2006) it is clear the concept for the wide availability of pyruvic acid in the central metabolism of plants is not introduced by them but actually by Eisenreich et al., Trends in Plant Sciences 2001, 7, 78-84, DOI: 10.1016/S1360-1385(00)01812-4. Similarly, the source for Jardine et al. (2010) could be checked.

We have added the Eisenreich et al. reference.

Pyruvic acid ($CH_3C(O)C(O)OH$), the simplest $\alpha$-keto-acid, is omnipresent in plants where it is central to the metabolism of isoprene, monoterpenes and sesquiterpenes (Eisenreich et al., 2001; Magel et al., 2006; Jardine et al., 2010).

In addition, in p. 2 l. 13 and l. 15, there have been four recent reports for the heterogeneous photooxidation of aromatics emitted during biomass burning that produced oxocarboxylic acids, including pyruvic acid by Pillar et al., which should be briefly described here: i) Pillar et al., Environmental Science & Technology 2017, 51, 951-4959, DOI: 10.1021/acs.est.7b00232; ii) Pillar et al., Environmental Science & Technology 2014, 48, 14352-14360. DOI: 10.1021/es504094x; iii) Pillar-Little et al., Environments 2018, 5, 104. DOI: 10.3390/environments5090104; and iv) Pillar et al., Journal of Physical Chemistry A 2015, 41,10349-10359. DOI: 10.1021/acs.jpca.5b07914.

We have modified the text. We now separate gas-phae and condensed-phase processes and include the formation of pyruvic acid (and 3-oxo-propionic acid) in the oxidation of aromatics formed during biomass burning.

There are several known routes to the photochemical formation of gas-phase pyruvic acid in the troposphere. In clean air, pyruvic acid is generated during the photo-oxidation of isoprene via the ozonolysis of methylvinylketone (MVK) and subsequent hydrolysis of the Criegee intermediates formed (Jacob and Wofsy, 1988; Grosjean et al., 1993; Paulot et al., 2009). Pyruvic acid is found in the photolysis (in air) of methylglyoxal (Raber and Moortgat, 1995), itself formed from the OH-initiated oxidation of several biogenic VOCs (Arey et al., 2009; Obermeyer et al., 2009) including monoterpenes (Yu et al., 1998; Fick et al., 2003). Pyruvic acid is also formed in the reactions of peroxy radicals generated in the oxidation of propane, acetone and hydroxyacetone (Jenkin et al., 1993; Warneck, 2005) and in the gas-phase photo-oxidation of aromatics in the presence of NOX (Grosjean, 1984; Praplan et al., 2014).

In the condensed phase, the aqueous-phase oxidation of methylglyoxal leads to the formation of pyruvic acid (Stefan and Bolton, 1999). Biomass burning also results in the formation of pyruvic acid (Andreae et al., 1987; Helas et al., 1992) where the heterogeneous photooxidation of particle-phase aromatics plays a role (Pillar et al., 2014; Pillar et al., 2015; Pillar and Guzman, 2017; Pillar-Little and Guzman, 2018). The latter also results in formation of a further oxo-carboxylic acid with the same mass (3-oxo-propionic acid). Finally, pyruvic acid is believed to be directly emitted by vegetation as indicated by measurements of very high mixing ratios under oxidation-free conditions in a tropical rain-forest biome (Jardine et al., 2010).

3) P. 2 l. 21-23 and p. 8 l. 25: There are some differences with more current literature than the cited, which need to be considered here and estimated during a comparison of phase processes and drivers for pyruvic acid loss in the conclusions sections of the manuscript. Particularly, the photochemical loss of pyruvic acid in water was evaluated by Eugene and Guzman, J. Phys. Chem. A 2017, 121, 15, 2924-2935, DOI: 10.1021/acs.jpca.6b11916, to occur with a lifetime of only ~22 min, and was indicated to vary by ~2-times in the gas phase.

This section (title: Atmospheric sinks of gas-phase pyruvic acid) focuses on the gas-phase losses of pyruvic acid. As the reviewer states, the rate of irreversible loss of pyruvic acid to the condensed-phase will change the overall loss rate constant for gas-phase pyruvic acid.

In the revised version we consider (section 3.1) the heterogeneous loss of gas-phase pyruvic acid to available aerosol in more detail when deriving a gas-phase lifetime (and source strength).

We also consider the loss of pyruvic acid via heterogeneous uptake to particles, which can be assessed via Eq. (1).

$$k_{het} = \frac{\gamma \, \bar{c} \, A}{4} \tag{1}$$

where $\gamma$ is the uptake coefficient, $A$ the aerosol surface area density (in cm$^2$ cm$^{-3}$), $\bar{c}$ the average thermal velocity (in cm s$^{-1}$).

Using, the mean aerosol surface area observed during IBAIRN of $2 \times 10^{-7}$ cm$^2$ cm$^{-3}$ (Liebmann et al., 2019), with $\bar{c} = 2.65 \times 10^4$ cm s$^{-1}$ at 290 K and an uptake coefficient of 0.06 reported for the uptake to aqueous surfaces (Eugene et al., 2018) we derive $k_{het} = 8 \times 10^{-5}$ s$^{-1}$.

The overall loss rate (photolysis + deposition heterogeneous loss) of pyruvic acid is then $L_{pyr} = 2.8 \times 10^{-4}$ s$^{-1}$, corresponding to a lifetime of $\approx$ 1h. We emphasise that the calculated lifetime (and thus the source strength we derive below) are very sensitive to the estimated deposition rate and are thus subject to major uncertainties. In addition, the appropriate uptake coefficient may be less than its value on pure water if the aerosol contains a large mass fraction of organic material which will reduce the rate of accommodation of pyruvic acid at the surface as has been seen for other trace gases e.g. N$_2$O$_5$ (Folkers et al., 2003; Abbatt et al., 2012).

Similarly, the lifetime for the loss of pyruvic acid reacting with OH radical in water was evaluated to be 1-10 days and in the gas phase to be 2-months.

While the rate constant for loss of aqueous phase pyruvic acid via reaction with OH is of interest in understanding the overall fate and role of pyruvic acid, it does not impact on the photochemical fate of (and production of gas-phase radicals from) gas-phase pyruvic acid, which are the foci of this manuscript.

4) P. 2 l. 28-30: It would be important to indicate here that one of the factors that enhances the solubility of pyruvic acid, as indicated to be "highly soluble" was explained in a paper by Guzman et al., J. Am. Chem. Soc. 2006, 128, 10621-10624, DOI: 10.1021/ja062039v; in terms of a cooperative reversible equilibrium of hydration that converts the carbonyl group into a geminal diol.

We now cite the Guzman et al paper on the underlying reason for the high solubility of pyruvic acid:

With a Henry's law solubility of $\approx 3 \times 10^5$ M atm$^{-1}$ (Staudinger and Roberts, 1996), pyruvic acid is highly soluble (due to formation of a germinal diol, (Guzman et al., 2006)) and wet and dry deposition and partitioning.

Similarly, for the mention of pyruvic acid "partitioning into the aerosol-phase" there should be a description here of recent work discussing the crystal structure of pyruvic acid by Heger et al., Acta Crystalographica E 2019, 75 (6), 858-862, DOI: 10.1107/S2056989019007072, and the novel study of the partitioning of gas phase pyruvic acid into water by Eugene et al., Langmuir 2018, 34, 9307-9313, DOI: 10.1021/acs.langmuir.8b01606. The last Langmuir paper should also be connected to the text about water clusters of pyruvic acid and pyruvate in p. 4 l. 26, and to the text in p. 8 l. 19-21 by including the reactive uptake coefficient of gaseous pyruvic acid by water reported to be 0.06.

We have modified the text on the loss of pyruvic acid to include heterogeneous uptake.

We also consider the loss of pyruvic acid via heterogeneous uptake to particles, which can be assessed via:

eq. 1

$$k_{\text{het}} = \frac{\gamma \, \bar{c} \, A}{4}$$
(1)

Where $\gamma$ is the uptake coefficient, $A$ the aerosol surface area density (in cm$^2$ cm$^{-3}$), $\bar{c}$ the average thermal velocity (in cm s$^{-1}$). Using, the mean aerosol surface area observed during IBAIRN of $2 \times 10^{-7}$ cm$^2$ cm$^{-3}$ (Liebmann et al., 2019), with $\bar{c} = 2.65 \times 10^4$ cm s$^{-1}$ at 290 K and an uptake coefficient of 0.06 reported for the uptake to aqueous surfaces (Eugene et al., 2018) we derive $k_{\text{het}} = 8 \times 10^{-5}$ s$^{-1}$.

The overall loss rate (photolysis + deposition  heterogeneous loss) of pyruvic acid is then $L_{\text{pyr}} = 2.8 \times 10^{-4}$ s$^{-1}$, corresponding to a lifetime of $\approx$ 1h. We emphasise that the calculated lifetime (and thus the source strength we derive below) are very sensitive to the estimated deposition rate and are thus subject to major uncertainties. In addition, the appropriate uptake coefficient may be less than its value on pure water if the aerosol contains a large mass fraction of organic material which will reduce the rate of accommodation of pyruvic acid at the surface as has been seen for other trace gases e.g. $N_2O_5$ (Folkers et al., 2003; Abbatt et al., 2012).

Finally, connected to the contribution of pyruvic acid to "SOA formation", it would be appropriate to include original related literature and the most updated papers:

i)      Guzman et al., J. Phys. Chem. A 2006, 110, 3619-3626, DOI: 10.1021/jp056097z;
ii)     Guzman et al., J. Phys. Chem. A 2006, 110, 931-935, DOI: 10.1021/jp053449t;
iii)    Xia et al., J. Phys. Chem. A 2018, 122, 6457- 6466, DOI: 10.1021/acs.jpca.8b05724,
iv)     Eugene and Guzman, Molecules 2019, 24, 1124; DOI: 10.3390/molecules24061124 et al.,
v)      Eugene and Guzman, Environ. Sci. Technol. 2019, DOI: 10.1021/acs.est.9b03742;
vi)     Mekic at al., Environ. Sci. Technol. 2018, 52, 12306- 12315, DOI: 10.1021/acs.est.8b03196; and
vii)    Mekic et al., Atmos. Environ. 2019, 219, 117046, DOI: 10.1016/j.atmosenv.2019.117046.

We already provide six references (between 2006 and 2017) that indicate that pyruvic acid is implicated in SOA formation. We would like to emphasise that the role of pyruvic acid in forming SOA is not the central theme of this manuscript and is only mentioned to give breadth to the role of pyruvic acid in the atmosphere. We see little gain in extending our list by a further 7 references, 5 of which are from the same research group. To update our reference list we now cite (additionally) the 2019 publication of Mekic et al., and Eugene et al.

…..thus contributing to SOA formation (Carlton et al., 2006; Tan et al., 2012; Griffith et al., 2013; Reed Harris et al., 2014; Eugene and Guzman, 2017; Eugene and Guzman, 2019; Mekic et al., 2019)

5) P. 4 l. 15: In standard analytical chemistry work the limit of detection (LOD) is established with a 99% chance of being greater than a blank for three times the standard deviation. It is confusing here that a value of 15 pptv is provided with 2 sigma for 10 s. This issue requires clarification.

Detection limits for trace-gases are often quoted at either $2\sigma$ or $3\sigma$. We prefer to stick to $2\sigma$ as in our previous publication describing this instrument (Eger, P. G., Helleis, F., Schuster, G., Phillips, G. J., Lelieveld, J., and Crowley, J. N.: Chemical ionization quadrupole mass spectrometer with an electrical discharge ion source for atmospheric trace gas measurement, Atmos. Meas. Tech., 12, 1935-1954, doi:10.5194/amt-12-1935-2019, 2019.)

6) P. 4 l. 23-24: More details about how the oxidation was accomplished in this work (and not in the reference cited) should be provided here. An associated calibration curve should be added to the supplementary information.

We now give the most important details.

…an infrared absorption spectrometer measuring $CO_2$ (LI-COR) following the complete oxidation of pyruvic acid to 3 $CO_2$ molecules in air, using a palladium catalyst at 350 °C (Veres et al., 2010)….

and provide a plot of the calibration curve in the supplementary information

This is now provided in Fig. S1 of the supplementary information.

7) P. 5 l. 4-5: This statement is unclear. What is the meaning of a 30 % ± 10 pptv mixing ratio? Clarify the units.

The uncertainty consists of a relative term (30 %) and a constant term (10 pptv). We now write:

We have increased the total uncertainty on the pyruvic acid mixing ratios to 30% of the mixing ratio ± 20 pptv to take account of this.

8) From p. 5 l. 22 to p. 6 l. 14, and Figure S1: The general idea and the analysis are acceptable but do not explain that the anion at m/z 87 could also be assigned to 3-oxopropanoic acid.

We are unaware of measurements of gas-phase 3-oxopropanoic acid which has been associated with SOA and biomass burning. During the IBAIRN campaign we were not impacted by biomass burning. However, we now mention that 3-oxopropanoic acid has the same chemical formula as pyruvic acid and could (in principal) contribute to the signal at m/z 87.

……the signals at m/z 214.921 ($C_3H_4O_3 \cdot I^-$) and m/z 87.008 ($C_3H_3O_3^-$) confirmed the assignment of m/z 87 to a molecule with three of each C- and O-atoms, and thus to pyruvic acid (2-oxo-propionic acid) or an isomer thereof such as 3-oxo-propionic acid, HC(O)CH$_2$C(O)OH (also known as formyl acetic acid or malonaldehydic acid).

…… We cannot rule out that 3-oxo propionic acid contributed to our CI-QMS signal at m/z 87 (or the HR-L-ToF-CIMS signal at m/z 87.008). However, as 3-oxo propionic acid has only been observed in the particle-phase and is associated with air masses impacted by biomass burning (Pillar and Guzman, 2017) our assumption that pyruvic acid is the dominant contributor to the signal at m/z 87 during IBAIRN appears justified.

9) P. 7 l. 21-30: This observation for the activity associated to the saw mill appears anecdotic in the context of evaluating pristine emissions from the natural boreal forest environment. The revisions should consider if the text is contradicting the objective of the manuscript.

We agree and have reduced the discussion related to the "saw-mill event" to a few lines.

During two periods of a few hours duration (9[th]-10th Sept), operations from a nearby saw-mill were apparent as elevated monoterpene mixing ratios (Eerdekens et al., 2009; Williams et al., 2011; Hakola et al., 2012). The influence of the saw-mill could be confirmed by examining 48 h back trajectories (HYSPLIT, Draxler and Rolph (2011). These periods are indicated in Fig. 1.

10) After the text is revised, and the calculations are rechecked based on the comments above and the inclusion of the reactive uptake coefficient of gaseous pyruvic acid by water in the model, some editions of section *3.2* will be needed in the revised manuscript.

The lifetime of pyruvic acid has been re-calculated and now includes heterogeneous uptake (see comment and reply above). The inclusion of heterogeneous loss of pyruvic acid reduces its lifetime from 1.4 to 1 hour. This impacts on its source strength (this has been recalculated, see above) but not on its mixing ratio in the gas-phase. Section 3.2 deals with the impact

(gas-phase only) of the measured mixing ratios of (gas-phase) pyruvic acid. We have changed the heading of this section to emphasise this:

3.2 Role of gas-phase pyruvic acid in the troposphere.

5 **Technical Corrections:**

P. 3. l. 20: Delete "but unexplored atmospheric component".
We deleted the words as suggested.

10 P. 4 l. 18-20: The statement "Laboratory … lower" is unnecessary and can be deleted.
We deleted this statement as suggested.

P. 9 l. 26: Add a mention to "Gas Phase".
We now write:
15 3.2 Role of gas-phase pyruvic acid in the troposphere.

P. 10 l. 3: Insert "gaseous" before pyruvic.
We now write:
Photolysis of gas-phase pyruvic acid.

P. 12 l. 9: add "in the gas phase" at the end.
20 We now write:
during pyruvic acid photolysis in the gas-phase.

---

## Author Comment (AC2) · 9 Dec 2019

**Reply to RC2**

5  *In the following, the referee's comments are reproduced (black) along with our replies (blue) and changes made to the text (red) in the revised manuscript.*

**General statement:**

10  This paper presents the first measurements of pyruvic acid in a boreal forest. There are pyruvic acid measurements in other locations but not many. In addition, pyruvic acid (unlike many other organic acids) has an active photochemistry and photolyzes rapidly. This allows it serve as a source of acetaldehyde which is probably the most important point in the paper. I think the results are interesting and probably should be published but I think the authors need to address some issues first. We thank the referee for the positive evaluation of our manuscript and the useful comments and suggestions. We have

15  modified the manuscript according to the comments listed below.

**Specific comments:**

1) I have some concerns about the measurement method. The largest is the use of an inlet that is heated to 200 °C. This is

20  done so the instrument can measure PANs (it would be nice to see these concentrations in the SI at least as well and may help rule out contributions to PiBN which could have a very large sensitivity). On page 5 we already rule out a contribution from PiBN (or PnBN) at $m/z$ 87 as the signal was not affected by addition of NO (which would prevent detection of both). We also note that the sensitivity to pyruvic acid, which we mention on page 4 (4.8 Hz pptv$^{-1}$ per $10^6$ Hz of I$^-$), is $\approx$ a factor 5 greater than for PAN (1.04 Hz pptv$^{-1}$ per $10^6$ Hz of I). The PAN dataset can be

25  found in Eger et al, 2019.

However, it may also lead to unwanted chemistry in the heated inlet. Do the authors know that the signal for pyruvic is present in the CIMS if the sampled air is unheated? Is the measured pyruvic sample modified by heating of the inlet? During the IBAIRN campaign the inlet as continuously heated. We found no significant change in the pyruvic acid signal

30  when operating the inlet "hot" or at room temperature during laboratory calibrations.

I assume the aerosol filter before the CIMS takes out all the organic aerosol that might evaporate. However, I would think it might significantly impact the transmission and time response to pyruvic. Was this tested? Could this filter significantly smear out the pyruvic observations? I would be especially concerned at high RH.

35  We agree that any inlet line and filter will serve to smear out the pyruvic acid signal over time scales of several minutes, thus precluding detection of very sharp concentration changes. As we see rapid concentration changes over time scales of 10 mins, we do not think that the stickiness of pyruvic acid will be large enough to change the diel profile of pyruvic acid and have added some text to mention this. Owing to its large affinity for surfaces, we expect that sharp changes in pyruvic acid mixing ratios (timescales of minutes)

40  will be smeared out over timescales of 10s of minutes because of adsorption and desorption on the inlet line and the filter and filter-holder. We do not expect that this will significantly impact the pyruvic acid time-series over the course of the diel cycle.

2) My second concern about the method is the measurement of the pyruvic background. In addition, I don't understand how

45  the LOD for the instrument is defined. I assume it depends on the variability of the background? So this should be clearly defined. I also don't understand how the background is measured. It seems like the steel wool scrubber doesn't react fast and so the authors chose not to use this as a background measurement? It is also stated that the background varies with the overall signal. OK if this is true then at least show this in the SI. There is also a mention that backgrounds are derived when the measurement is near LOD or when the instrument is overflowed with synthetic air. So I really don't understand how the

background signal is measured and interpolated as function of time. I think this needs to be clearly explained in detail with graphical examples at least given in the SI. I also don't understand how overflowing with zero air is considered preferable to scrubbing ambient air – I am sure it is easier to deal with but I am not sure it is valid. Especially if the zero air is a different RH or temperature than ambient. My guess is that pyruvic acid is similar to nitric acid and the whole key to the measurement may be accurately measuring the instrument background.

As the pyruvic signal did not go to zero (but tracked the ambient concentration) when scrubbing ambient air to obtain a background signal we could not use variability in the background signal to estimate the LOD. We have modified the text that describes how the background signal on $m/z$ 87 was obtained: The zero air was passed through the heated inlet and thus at the same temperature as ambient air when entering the IMR.

Owing to pyruvic acid's high affinity for surfaces, it typically took 5–10 minutes to remove > 90 % of the signal, which resulted in a background signal that co-varied with the ambient signal at $m/z$ 87. This is illustrated in Fig. S2 of the supplementary information in which we show the raw signal at $m/z$ 87 and the signal at the same mass when sampling via the scrubber. The background signal for $m/z$ 87 (red dashed line in Fig. S2) was therefore determined from measurements in which pyruvic acid mixing ratios were close to the detection limit during the early part of the campaign. This choice could be confirmed by examining the background signal when the inlet was overflowed with zero-air (red points in Fig S2.). We have increased the total uncertainty on the pyruvic acid mixing ratios to 30% (of the mixing ratio) ± 20 pptv to take account of this.

3) Did you measure formic or acetic acid during the campaign? Formic should be readily apparent at its cluster with I-. Is there evidence for acetic acid at 59?

Our instrument has very low sensitivity to organic acids as clusters with I⁻. This is related to collisional dissociation effects and was mentioned in our recent publication (Eger et al., 2019) describing this instrument. As described in Eger et al. (2019), we measure the sum of acetic acid and peracetic acid at $m/z$ 59.

The presence of butanoic acid is ruled out in part by assuming a similar sensitivity as acetic acid (i.e. lower than for pyruvic). Why not just measure the butanoic sensitivity?

We did not calibrate the QI-CIMS for detection of butanoic acid. Unfortunately, this cannot be done retrospectively as the instrument is no longer operated with the same ion-source. As stated in the text, enhanced levels of butanoic acid at this remote site are highly unlikely.

If there are significant levels or formic and acetic then you might expect other carboxylic acids as well. I also don't think you can argue that the lack of butanoic in the Matilla et al work argues that it is not present in this region. The pyruvic acid is certainly very different in many respects between the two locations if this is true.

We argue that butanoic acid (which is found in anthropogenically impacted air masses) is unlikely to be present at levels of > 2ppb (necessary to explain the $m/z$ 87 signal). We cannot rule out that butanoic acid will contribute to some extent to the $m/z$ 87 signal. We write:

Similar arguments help us to rule out a large contribution on $m/z$ 87 from butanoic acid, which accompanies anthropogenic activity (e.g. traffic emissions, see Mattila et al. (2018)) and would not acquire continuously high concentrations at this site. Assuming similar sensitivities for butanoic and acetic acid, i.e. 0.62 Hz pptv$^{-1}$ (Eger et al., 2019) butanoic acid mixing ratios sometimes exceeding 2.5 ppbv would be required to account for the entire signal at $m/z$ 87. In the absence of independent measurements of butanoic acid during IBAIRN, we can only conclude that it is unlikely to represent a significant fraction of the CI-QMS signal at $m/z$ 87.

4) The sensitivity is reported per MHz of I-. What was the typical sensitivity?

The sensitivy (i.e. ion count at $m/z$ 87 per ppt of pyruvic acid) depends on the concentration of primary ions (e.g. I-) in the ion-molecule reactor and must therefore be cited as a "normalised sensitivity" as we have done.

In addition, if I(CN)2- is the true reagent ion why report sensitivity relative to I-?

We suggest on page 4 that, analogous to acetic acid, I(CN)$_2^-$ may play a role in pyruvic acid detection with our electrical discharge ion source. We cannot prove this. However, as the C- and I-atoms both stem from CH$_3$I, we expect the

concentration of $I(CN)_2^-$ to be proportional to that of $I^-$, which (unlike $I(CN)_2^-$) was continuously monitored during the campaign and thus serves as the best primary ion to which we can normalise signal. We now write:

The $I(CN)_2^-$ ion was not monitored continuously during the IBAIRN campaign and the signal at *m/z* 87 was converted to a mixing ratio after normalisation to the ion count of the major primary ion, $I^-$. As the I and C atoms in $I(CN)_2^-$ stem from $CH_3I$, we expect the concentration of $I(CN)_2^-$ to be proportional to that of $I^-$ (which was monitored continuously).

How were post mission calibrations related to the field data?

We determined the sensitivity per ppt of pyruvic acid normalised to $I^-$, and determined that there was no significant humidity dependence. The text on page 4 has been modified:

As detection of pyruvic acid during IBAIRN was not expected, the instrument was calibrated post-campaign by simultaneously monitoring the output of a diffusion source (98 % pyruvic acid, Sigma-Aldrich) with the CI-QMS and an infrared absorption spectrometer measuring $CO_2$ (LI-COR) following the complete oxidation of pyruvic acid to 3 $CO_2$ molecules in air, using a paladium catalyst at 350 °C (Veres et al., 2010). A calibration curve is given in Fig S1 of the supplementary information. The CI-QMS sensitivity to pyruvic acid was found to be independent of relative humidity (RH) for RH > 20 %. In dry air the sensitivity drops to about 60 % of that observed with humidified air reflecting the importance of water clusters in the reaction with the primary ion.

5) The Jardine et al., 2010 observations were done in a biosphere and are not truly ambient and in my opinion are closer to a chamber experiment. So I would not classify them as ambient measurements.

The salient point is that the high mixing ratios of pyruvic acid reported by Jardine et al. were the result of emissions from the biosphere. Perhaps "enclosure" would be the better term (rather than chamber). We now write:

Mixing ratios of pyruvic acid (up to 15 ppbv) were reported in an experimental tropical rain forest enclosure (Jardine et al., 2010) and were accompanied by isoprene levels exceeding 100 ppbv with other terpenoids up to ~ 10 ppbv.

6) I don't understand including the contaminated "sawmill" days in the data. I think it only confuses things. I would remove from the dataset and concentrate on the clean boreal forest conditions.

We have removed the figure (2) that focussed on the saw-mill events and now only mention this in passing.

During two periods of a few hours duration (9[th]-10th Sept), operations from a nearby saw-mill were apparent as elevated monoterpene mixing ratios (Eerdekens et al., 2009; Williams et al., 2011; Hakola et al., 2012). The influence of the saw-mill could be confirmed by examining 48 h back trajectories (HYSPLIT, Draxler and Rolph (2011). These periods are indicated in Fig. 1.

7) I am not sure I agree with the following statement "On several days the pyruvic acid mixing ratios co-vary with those of isoprene and monoterpenes, with night-time maxima resulting from emissions into the very shallow boundary layer, which is especially apparent in the period 9–15 September 2016".

I certainly can't make it out from the way the data is presented. If this statement is to be maintained it needs to be backed up with a figure that shows it more clearly. If it co-varies some days why not others?

The use of the GC vs. PTRMS data for isoprene and monoterpenes also needs to be clarified as they look pretty different. Is there a measurement of boundary layer height to back up this statement?

We observe coincident maxima and minima in the mixing ratios of pyruvic acid, isoprene and monoterpenes. We have replaced the previous Figure 2 (saw-mill) with one that focusses on this period to illustrate this. Co-variance will be weakened if the sinks of pyruvic acid do not vary in the same way as those of isoprene and monoterpenes. As pyruvic acid lifetime will be influenced by dry deposition and uptake to aerosol (whereas the lifetimes of isoprene and monoterpenes are not), we expect loss of correlation if these processes are important. The day- night variation of the boundary layer height is already mentioned in Section 3. We have added a Figure (new Fig. 2) to make the co-variation of pyruvic acid with isoprene and mono-terpenes more apparent. The new text is:

During two periods of a few hours duration (9[th]-10th Sept), operations from a nearby saw-mill were apparent as elevated monoterpene mixing ratios (Eerdekens et al., 2009; Williams et al., 2011; Hakola et al., 2012). The influence of the saw-mill could be confirmed by examining 48 h back trajectories (HYSPLIT, Draxler and Rolph (2011). These periods are highlighted (grey shading) in Fig. 2 which focuses on a section of the campaign in which the pyruvic acid mixing ratios were

rather variable and we compare them with those of isoprene and monoterpenes. There is apparent co-variation of pyruvic acid with isoprene and monoterpenes, the night-time maxima resulting from emissions into the very shallow boundary layer. As we discuss later, the mixing ratios of highly soluble pyruvic acid will be more strongly influenced by deposition of scavenging by aqueous particles than isoprene or monoterpenes so that there is no reason to expect continuously high correlation between these trace-gases as meteorological conditions change.

Also I think Figure 3 is misleading. I am not convinced that you have a regular diurnal variation in pyruvic. There needs to be error bars in this figure that represents the deviation from average.
The point of this figure is not to show that there is a regular diel variation in pyruvic acid. The time series indicates that this is not the case. We use this median profile to make some calculations that are representative for a longer period and to indicate that pyruvic acid, isoprene and mono-terpenes have a common maximum at ~ 17:00 UTC.
We agree that the variability needs to be addressed and now show a plot in the SI with the 25 and 75% percentiles for each trace gas.
We have made a new Figure 3 with 25th and 75th percentiles as well as the median. We have separated this from (new) Fig. 4 which shows the medial diel profiles of OH, $O_3$ and $J_{pyr}$.

8) WRT the pyruvic observations I am most bothered/intrigued that it doesn't seem to go to zero at night for the latter part of the campaign. This is very different than the Matilla et al. work which shows that the pyruvic goes to zero at night and also is greatly diminished during the day at lower altitudes. This implies efficient dry deposition in Colorado that I think should be active in Finland as well. Given that the RH goes to 100% on several nights as well I would expect dew formation and even greater enhancements in the loss of pyruvic at night. This needs to be addressed by the authors. Do they think the loss of pyruvic to the forest at night is minimized? Do emissions need to go up at night to explain observations?
The boundary layer was very shallow at night (35 m) compared to day ( > 500m). For constant emission, even if the loss rate at night was a factor of 5 greater (i.e. due to uptake to surfaces), we would still have seen a nocturnal increase in the mixing ratio of pyruvic acid. Potentially great differences in boundary layer dynamics make comparison with Matilla et al difficult.

9) I am also struck by the lack of correlation of pyruvic with temperature, J, etc. If pyruvic is truly emitted by vegetation then I think at the least it should be related fairly strongly to temperature. Is there an example of plant metabolite emissions that are unrelated to temperature? or PAR?
The same statement could be made about monoterpenes at this site. The emission of monoterpenes is stronger during the (warm) day, but the mixing ratios are higher at night. This is caused by boundary layer height variations (see above).

10) In summary, I think the authors have demonstrated that the observed pyruvic levels are not due to isoprene or terpene photochemistry. However, I am not totally convinced they are observing only pyruvic acid. I am also skeptical that vegetation would emit pyruvic in a manner needed to explain the observations. So I do think that the authors need to restate their conclusions (i.e. in abstract) especially since they have not demonstrated a flux of pyruvic to the atmosphere. I agree that further measurements including flux and altitude profiles would be very useful to sort this out.
We are not the first to suggest that pyruvic acid is directly emitted by vegetation. The enclosure experiments of Jardine et al strongly indicate that this is the case. As we mention in the text, we cannot exclude contributions from other trace gases, but suggest that pyruvic acid is the dominant source of signal at $m/z$ 87. As we indicate in the conclusions, further measurements e.g. using high resolution mass-spectrometry and chromatographic separation methods would be useful in confirming the results of this study.

It would be very interesting if the boreal forest emits as much pyruvic acids as monoterpenes to the atmosphere.
Indeed. As we conclude, the dependence of pyruvic acid emissions on temperature, season etc. require more studies using specific analytical methods.

---

## Referee Report (RR1)

I appreciate the author's response to my review. I suggest publication of the paper after consideration of some minor points below.

1) I am still concerned that the measurements were made with an inlet heated to 200 C. The authors state that this did not impact lab calibrations. I would not expect them too. I would be concerned that heating the inlet leads to secondary chemistry that impacts their pyruvic signal under ambient conditions in the boreal forest. I agree this will never be sorted out since they didn't test this in the field but I think this needs to be mentioned as a potential issue.

2) I understand much better how the background was determined. However, I don't agree with ignoring the results from the scrubber. In fact, I would expect the instrument background of a condensable substance to behave this way. So I think you should subtract the measured background. I think the zero air background is largely meaningless. The authors do put a large error bar on the measurement to account for the background. However, I think they need to take out the statement that they it is inappropriate to use the scrubbed background.

3) I don't understand why you can't report a typical sensitivity. I understand you are normalizing your signal (even though you may not have measured all your effective reagent ions) to the I- signal. So I still think it is worth reporting your sensitivity at your average level of reagent ion.

4)

The use of the GC vs. PTR-MS data for isoprene and monoterpenes still needs to be clarified as they look pretty different. This is an important point as I think the GC data should be more immune to interferences than the PTR data. So why leave this off Figure 2? Why not show both? Which dataset is more applicable? There needs to be some discussion of this issue.

5) I am still bothered that the pyruvic never goes to zero and is often maximum at night. I understand the monoterpenes can do the same at night as well. This is partly due the sink not being as large as night as well as the collapse of the boundary layer. However, the monoterpenes are not soluble. So I think the striking difference between the Matilla et al work and the boreal results in this paper at least needs to be acknowledged and discussed to some extent. The Farmer group data was a very nice examination of both the altitude and diurnal profile of pyruvic. Their results are inconsistent with the results in this work. Perhaps it is due to the differences in emission rates due to vegetation etc. (plausible) or very different boundary layer dynamics (hard to believe). So not discussing the comparison of the measurements is a detriment to this paper.

---

## Author Response (AR2)

**Reply to RC2, re-review**

5 *In the following, the referee's comments are reproduced (green highlighting) along with our replies (blue) and changes made to the text (red) in the revised manuscript.*

10 1) I am still concerned that the measurements were made with an inlet heated to 200 C. The authors state that this did not impact lab calibrations. I would not expect them too. I would be concerned that heating the inlet leads to secondary chemistry that impacts their pyruvic signal under ambient conditions in the boreal forest. I agree this will never be sorted out since they didn't test this in the field but I think this needs to be mentioned as a potential issue.

15 We do not think that this is a significant problem and have added the following text: We cannot completely rule out that some unknown secondary reactions at 200 °C may influence the pyruvic acid concentration during ambient measurements, though, given the short inlet residence time (200 ms), we consider this to be unlikely.

20 2) I understand much better how the background was determined. However, I don't agree with ignoring the results from the scrubber. In fact, I would expect the instrument background of a condensable substance to behave this way. So I think you should subtract the measured background. I think the zero air background is largely meaningless. The authors do put a large error bar on the measurement to account for the background. However, I think they need to take out the statement that they it is inappropriate to use the scrubbed 25 background.

We disagree. Perhaps the important point to make (we had omitted to mention this before) is that the scrubbed background changes (gets smaller) with time. This is now illustrated in Fig. S2. If we had scrubbed for a longer period, the scrubbed background would have decreased towards the true instrument background level. The signal we observe when scrubbing for 30 short periods is the result of desorption of pyruvic acid from surfaces that, in terms of flow, are behind the scrubber. As atmospheric concentrations of pyruvic acid do not change as rapidly as the changes induced by scrubbing, disregarding the (average) background during from incomplete scrubbing is the correct approach. The following text has been added and Fig S2 modified:
Owing to pyruvic acid's high affinity for surfaces, even after 5–10 minutes of scrubbing, the signal did not go to zero (Fig. 35 S2), which resulted in a background signal that co-varied with the ambient signal at $m/z$ 87. This is illustrated in Fig. S2 of the supplementary information in which we show the raw signal at $m/z$ 87 and the signal at the same mass when sampling via the scrubber.

[Figure]

3) I don't understand why you can't report a typical sensitivity. I understand you are normalizing your signal (even though you may not have measured all your effective reagent ions) to the I- signal. So I still think it is worth reporting your sensitivity at your average level of reagent ion.

We now report a typical sensitivity for a typical primary ion count rate. The sensitivity was 4.8 Hz per pptv of pyruvic acid for a (typical) primary-ion count rate (at $m/z = 127$) of ~$10^6$ Hz.

4) The use of the GC vs. PTR-MS data for isoprene and monoterpenes still needs to be clarified as they look pretty different. This is an important point as I think the GC data should be more immune to interferences than the PTR data. So why leave this off Figure 2? Why not show both? Which dataset is more applicable? There needs to be some discussion of this issue.

We agree that the agreement between the GC-AED and PTR datasets for isoprene is poor. On the other hand the monoterpene datasets are in reasonably agreement. We have added the GC measurements to Figure 2.

We have to bear in mind that, as already stated, the instruments were not co-located but were 170 m apart. The GC sampled from a clearing, the PTR in denser forest. In a previous article describing results from IBAIRN we have highlighted the differences between measurements of terpenoids by these instruments, which is related to the inhomogeneous emission of isoprene and monoterpenes. As already stated, the choice of PTR over GC was guided mainly by the poor time resolution of the GC. We have added text stating that high time resolution, co-located measurements of terpenoids and pyruvic acid are necessary: Further field and enclosure studies are necessary to quantify its emissions and role during other seasons and to better understand its sources and sinks (e.g. generation in $OH/O_3/NO_3$ initiated oxidation of terpenes and dry deposition rates) in the boreal forest as well as in other environments. To this end, co-located, high-time-resolution measurements of mixing ratios and fluxes of terpenoids and pyruvic acid are necessary.

5) I am still bothered that the pyruvic never goes to zero and is often maximum at night. I understand the monoterpenes can do the same at night as well. This is partly due the sink not being as large as night as well as the collapse of the boundary layer. However, the monoterpenes are not soluble. So I think the striking difference between the Matilla et al work and the boreal results in this paper at least needs to be acknowledged and discussed to some extent. The Farmer group data was a very nice examination of both the altitude and diurnal profile of pyruvic. Their results are inconsistent with the results in this work. Perhaps it is due to the differences in emission rates due to vegetation etc. (plausible) or very different boundary layer dynamics (hard to believe). So not discussing the comparison of the measurements is a detriment to this paper.

We agree that the results of Mattila and ours are divergent. We question whether similar sources and sinks of pyruvic acid are likely to apply when comparing data obtained in the autumn in the middle of the boreal forest and in the summer in the Colorado Front Range (which, according to Mattila is influenced by highway traffic, oil and natural gas wells and animal feeding operations and where "……vegetation in the region is sparse, particularly during the hot, dry summer".

As requested, we have added some text describing the Mattila results in more detail and highlighting the differences in the observations and conclusions drawn by these two studies: In contrast, Mattila et al (2018) provide convincing evidence for photochemical production of pyruvic acid resulting in a mean mixing-ratio of 180 pptv maximising with photochemical activity. Mattila et al. (2018) also observed a strong reduction in the mixing ratio of pyruvic acid with height within the boundary layer due to dry deposition and found no evidence for strong surface emissions. As described above, dry deposition will also have impacted on the pyruvic acid mixing ratios observed at Hyytiälä, though in the absence of vertical profiles or flux measurements it is difficult to assess rigorously its impact during day or night. The differences between the summertime measurements of Mattila et al. (2018) and the present work are very likely related to the starkly contrasting environments: The IBAIRN campaign being conducted in the remote, boreal forest in Autumn whereas Mattila et al (2018) made their

summertime measurements over an eight-day period in a region with sparse vegetation and with significant anthropogenic influence from traffic, oil and natural gas operations and livestock and $NO_X$ levels of several ppbv, and concluded that pyruvic acid was generated photochemically from aromatics emitted by automobiles.

[revised manuscript text omitted]

**Figure S2:** *Upper panel:* Pyruvic acid mixing ratio (measured at *m/z* 87) by CI-QMS during IBAIRN. The ambient air signal (converted to mixing ratio) is represented by the black curve, the background obtained by bypassing the air through the scrubber is illustrated by the blue curve. *Lower panel:* Raw data showing measurement (black) and scrubbing periods (blue). During scrubbing, the signal decays quasi-exponentially towards the red dotted line, which is the subtracted background signal, measured by overflowing the inlet with zero-air.

[Figure]

**Figure S3:** Signals measured by the HR-L-ToF-CIMS at *m/z* 87.008 (assigned to pyruvic acid) and 87.045 (assigned to butanoic acid).

[Figure]

**Figure S4:** Acetaldehyde ($CH_3CHO$) production rates over the diel cycle based on measurements (median diel profiles for IBAIRN) of pyruvic acid and $J_{pyr}$ (assuming a photolysis yield of $\varphi = 0.2$), calculated OH and estimated mixing ratios of alkanes from literature data (see manuscript). The dashed vertical line indicates solar noon.